



# Demystifying academics to enhance university-business collaborations in environmental science

[1]John K. Hillier, [2]Geoffrey Saville, [3]Mike J. Smith, [4]Alister J. Scott, [5]Emma K. Raven, [2]Jonathan Gascoigne, [1]Louise Slater, [6]Nevil Quinn, [7]Andreas Tsanakas, [8]Claire Souch, [9]Gregor C. Leckebusch, [10]Neil Macdonald, [11]Jennifer Loxton, [12]Rebecca Wilebore, [13]Alexandra Collins, [14]Colin MacKechnie, [15]Jaqui Tweddle, [16]Alice M. Milner, [17]Sarah Moller, [18]MacKenzie Dove, [19]Harry Langford, [20]Jim Craig

[1]Geography and Environment, Loughbrough University, Loughborough, LE11 3TU, UK.
[2]Willis Towers Watson, 51 Lime Street, London, England EC3M 7DQ, UK.
[3]School of Geography, Earth and Environmental Sciences, University of Plymouth, Plymouth, PL4 8AA, UK.
[4]Dept. Geography and Environmental Sciences, Northumbria University, Newcastle, UK.
[5]JBA Risk Management, South Barn, Broughton Hall, Skipton, North Yorkshire, BD23 3AE, UK.
[6]Dept. Geography and Environmental Management, University of the West of England, Bristol, BS16 1QY, UK.
[7]Cass Business School, City, University of London, 106 Bunhill Row, London EC1Y 8TZ, UK.
[8]AWHA Consulting, 67 Worcester Point, Central Street, London, EC1V 8AZ, UK.
[9]School of Geography, Earth and Environmental Sciences, University of Birmingham, Birmingham, UK.
[10]School of Environmental Sciences, University of Liverpool, Liverpool, L69 7ZT, UK.
[11]School of Geosciences, University of Edinburgh, Edinburgh, EH93JW, UK.
[12]Dept. Zoology, University of Oxford, Oxford, OX1 3SZ, UK.
[13]Centre for Environmental Policy, Faculty of Natural Sciences, Imperial College London, London, SW7 2AZ, UK.
[14]Centre for Ecology & Hydrology, Wallingford, OX10 8BB, UK.
[15]School of Biological Sciences, University of Aberdeen, Aberdeen, AB24 2TZ, UK.
[16]Department of Geography, Royal Holloway, University of London, Egham, TW20 0EX, UK.
[17]Dept. Chemistry, University of York, York, YO10 5DD, UK.
[18]Walker Institute, University of Reading, UK.
[19]Dept. Geography, University of Sheffield, Sheffield, S10 2TN, UK.
[20]OasisHub, 40 Bermondsey Street, London, SE1 3UD, UK.

*Correspondence to*: John Hillier (j.hillier@lboro.ac.uk)

**Abstract.** In countries globally (e.g. UK, Australia) there is intense political interest in fostering effective university-business collaborations, but there has been scant attention devoted to exactly *how* individual scientists' workload (i.e. specified tasks) and incentive structures (i.e. assessment criteria) may act as a key barrier to this. To investigate this an original, empirical dataset is derived from UK job specifications and promotion criteria, which distil universities' varied drivers into requirements upon academics. This reveals the nature of the severe challenge posed by a heavily time-constrained culture; specifically, a tension exists between opportunities presented by working with industry and non-optional duties (e.g. administration, teaching). Thus, to justify the time to work with industry, such work must inspire curiosity and facilitate future novel science in order to mitigate its conflict with the overriding imperative for academics to publish. It must also provide evidence of real-world changes (i.e. impact), and ideally other reportable outcomes (e.g. official status as a business' advisor), to feed back into the scientist's performance appraisals. Indicatively, amid 20-50 key duties, scientists





may be able to free *up to* 0.5 days/week for work with industry. Thus specific, pragmatic actions, including short-term and time-efficient steps, are proposed in a 'user guide' to help initiate and nurture a long-term collaboration between an early- to mid-career environmental scientist and a practitioner in the insurance industry. These actions are mapped back to a tailored typology of impact and newly-created representative set of appraisal criteria to explain *how* they may be effective, mutually

beneficial, and overcome barriers. Throughout, the focus is on environmental science, with illustrative detail provided through the example of natural hazard risk modelling in the insurance industry. However, a new conceptual model is developed, joining perspectives from literatures on academics' motivations and performance assessment, which we tentatively posit is widely applicable. Sector-specific details (e.g. list of relevant impacts, 'user guide') may serve as templates globally and across sectors.

**Key words**: University-business collaboration, impact, innovation, knowledge exchange, job specification, appraisal criteria, risk practitioner, catastrophe modelling, insurance sector.

## 1 Introduction

Political interest is increasing in converting research excellence into commercial success (Dowling, 2015; Evans, 2016; e.g.

Mowrey and Nelson, 2004) and societal impact (e.g. Reed, 2018). Thus, the idea of the 'entrepreneurial university' is popular (Etzkowitz, 2003; e.g. Slaughter and Leslie, 1997); it is argued both that universities might be fundamentally transforming into engines of economic growth (e.g. Feller, 1990; Florida and Cohen, 1999), or that there is a convergence to a hybrid where differences between scholarly and industrial activity become blurred (e.g. Owen-Smith, 2003). However, university-business collaborations could produce better outcomes through improved flow (a.k.a 'diffusion') of science innovation into

policy and decision-making practice (e.g. Dowling, 2015). This applies even in nations (e.g. UK, Australia) that rank relatively highly in the 'Global Innovation Index' (Dowling, 2015; Dutta et al., 2017; Evans, 2016). So, debate continues about how to incentivize, deliver, monitor, and support such a change. This, and a political desire to see collaborations be more productive, is attested to by 14 reviews and studies in the UK on this topic in the last 12 years (see Dowling, 2015).

In the insurance sector, the flow of university-produced environmental science into natural hazard risk assessment models (e.g. flood, earthquake, tropical cyclone), and thus associated decision-making, is substantial yet imperfect. These 'Catastrophe Models' (see Mitchell-Wallace et al., 2017) are vital in defining and implementing the financial mechanisms (e.g. re-insurance, catastrophe bonds) used to build resilience to these large risks; illustratively, $386 billion damage accrued to insured assets alone in 2011 (Von Peter et al., 2012). However, inter-model differences exist (e.g. $13-72 billion for

hurricane Maria in 1997 (KCC, 2018)). Thus, there is significant commercial interest in implementing the latest science to build the most realistic risk models. For example, the tendency of extra-tropical cyclones impacting Europe to cluster in time (i.e. occur in groups) (e.g. Vitolo et al., 2009) has been included, and better understanding the tentative indicators that flood

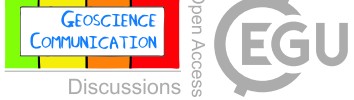



and wind damage tend to co-occur (e.g. Hillier et al., 2015; De Luca et al., 2017) is ranked as a current priority for this peril by insurers in a survey for the Lighthill Risk Network (Dixon et al., 2017).

In short, both industry and the university sector are seeking effective (i.e. mutually beneficial) pathways from science production in universities to pragmatic implementation, overcoming barriers.

The insurance sector (see Ch. 2.3 of Mitchell-Wallace et al., 2017) consists of entities that hold risk themselves (i.e. insurers, reinsurers and other financial institutions), companies who provide tools or advice to help them do so effectively (i.e. brokers, consultants, 'vendor' model companies), and regulators. With a few exceptions (e.g. Collette et al., 2010), much of

the research and development to create risk models (e.g. RMS, AIR, SwissRe, JBA Risk Management) is undertaken in-house and typically applies selected knowledge from previously published peer-reviewed research, rather than collaboratively generating new knowledge. Quantifying benefits of direct collaborations with university-based scientists (e.g. to make a business case) is non-trivial; this is particularly pertinent as a company might have >50% of technical staff qualified at MSc or PhD level. Thus, a partial barrier to co-working and knowledge exchange (KE) directly between

university scientists and practitioners evidently exists, although its origins may be complex. More widely, in practice few (re)insurers have the ability to directly approach academia to question choices made about research applied in models or to keep abreast of the latest findings. By better understanding what motivates academic researchers, it should be possible for industry to forge synergies more readily with world leading scientists. This will assist new and existing (e.g. Willis' Research Network and AXA's research fund) initiatives as they strive to most effectively access cutting-edge research to drive

innovation.

Challenges to collaboration vary by stakeholder (e.g. university, academic, industry) (e.g. Abreu et al., 2009; Dowling, 2015). For instance, business-relevant questions can be listed by insurers (Dixon et al., 2017; Lighthill Risk Network, 2016), but it can be hard to translate industrial needs into research questions that are precise enough for scientists to be *able* to

answer and intriguing and novel enough for scientists to *want* to prioritise answering them.

Thus, effective university-business collaboration requires mutual understanding (e.g. Dowling, 2015), and developing this demands investment of time and effort. Scientists would benefit from a greater appreciation of business drivers, needs and constraints, and we propose that industry (e.g. insurers) would be aided by understanding the answer to two questions: What

motivates academics to do specific work? And, reciprocally, what might constrain them? By demystifying the motives of university scientists, this paper aims to make it easier to develop collaborations that are feasible and produce timely outputs, perhaps to inform model creation and decision-making in insurance.



In academic debate, models (e.g. 'diffusion' (e.g. Rogers, 2003; Scott et al., 2018; Ward et al., 2009)) are used to understand how science may be deployed in business and the cultural, institutional and individual barriers to this, but there has been limited attention devoted to the exact nature of barriers facing academics, motivations to surmount them, and coping strategies to do so.

Like industry, university science is a complex landscape and the views, requirements and motivations of its actors (e.g. universities, individual academics, funding bodies) are not homogenous (Evans, 2016; e.g. Lam, 2011).   Conventional wisdom suggests that Intellectual Property (IP) and cultural differences are key barriers to collaboration (Abreu et al., 2009; Lambert, 2003), and this is still borne out to some extent by studies such as Dowling (2015) that consulted a variety of

stakeholders (e.g. universities, SMEs, Trade Associations) in which only 10-15% of the input was from scientists themselves. Studies that consulted only university-based scientists as individuals (24,443 respondents in total), however, disagree strongly and rate these factors as relatively unimportant (Abreu et al., 2009; D'Este and Perkmann, 2011; Evans, 2016). These place limitations on time in a scientist's working day as an important (Evans, 2016; e.g. Lazarsfeld-Jensen and Morgan, 2009), and perhaps the overriding (Abreu et al., 2009), constraint on university-industry collaboration. This will

likely therefore dominate an academic's decision making, framing actions whatever their motivations and desires may or may not be.

Conversely, it is possible to be positive and look at academics' motivations. D'Este and Perkmann (2011) review the recent literature on university-industry interaction including both informal (i.e. collaborative) modes (Grimple and Fier, 2010; Link

et al., 2007) and more heavily studied routes (i.e. patenting, licensing, spin-off companies) (Bercovitz and Feldman, 2006; e.g. Carayol, 2003; McMillan Group, 2016). Collaboration is the most frequent channel for interaction (D'Este and Patel, 2007; Perkmann and Walsh, 2007) including joint 'pre-competitive' research that is often subsidized by public funding, heavily directed contract research of immediate industry relevance, and consulting. In such collaboration, a small number of studies describe *what* academics' motivations are, compiling sizeable lists (e.g. Dowling, 2015). Studies delving deeper to

understand *why* these motivations arise are rare. For most UK academics, the driving incentive to interact with industry is to further their research (D'Este and Perkmann, 2011). Lam (2011) divides motivations into , 'gold' (i.e. personal income), 'puzzle' (i.e. knowledge/curiosity) and 'ribbon' (i.e. funding/reputation) (Stephan and Levin, 1992), finding that a great majority of practicing university-based scientists are motivated by the latter two traditional rewards.

Part of generating viable industry-university collaboration is that strategic and policy-level drivers must align with the incentive structures and circumstances of individual scientists for work to actually proceed.  These, sometimes conflicting, motivators are the main subject of this paper. The study's novelty is 3-fold. Firstly, direct and innovative data collection methods (Sect. 2) allow a broadly based (i.e. multi-university) and yet detailed view; consequently, a first synthesis in this context of academics' day-to-day duties and career-defining aspirational targets can be provided (Sects. 4.2, 4.3). Secondly,



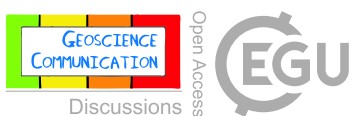

individual academics' performance evaluation has been only recently explicitly considered with respect to the research-teaching dipole (i.e. Cadez et al., 2017; Harland and Wald, 2018), and it fills a research gap by incorporating impact to this tensioned relationship. By investigating the day-to-day demands (i.e. micro-politics (e.g. McAreavey, 2006)) upon an illustrative scientist it gives insights into the pressures on their time (Sect. 5.1), and uses aspirational targets that govern their

appraisal (Sect. 5.2), to add detailed and diagnostic understanding of *why* actions are prioritised. A conceptual model is developed (Fig. 3). Thirdly, pragmatic suggestions for specific actions to initiate and nurture a collaboration are proposed (Sect. 5.3). These are mapped back to appraisal criteria (Table 2) and impact typology (Sect. 5.2.2) (Reed, 2018) to explain *why* actions may be effective, mutually-beneficial, and overcome the barriers that may be deterring scientists from working with industry.

This work is differentiated by framing it for an industry practitioner who engages with environmental science. Namely, the *what* and *why* that motivate university scientists are presented with the ultimate aim of conveying *how* a scientist might be pragmatically supported to effectively initiate and nurture collaborations with an industry practitioner to the mutual benefit of both parties: in other words, to provide a 'user guide' for practitioners. To this end, theory (Amsterdamska, 1990; e.g.

Latour, 1987; Rogers, 2003) is kept brief, and specific examples are favoured over generalities where possible. One way it focuses is by only considering the varied (e.g. work-life balance, teaching, promotion) and multi-level (e.g. government, university) drivers as they affect the persona of a hypothetical illustrative environmental scientist of ≲10 years faculty experience at a UK university (see Sect. 3). A second way is limiting illustrative, sector specific detail to that for risk practitioners and reinsurance.

Data and analysis are based within the environmental science discipline, but aspects of the analysis may be applicable more widely (e.g. social science, engineering) with caution, and the practical guide for risk practitioners in insurance (i.e. Sect. 5.3) may serve as a template.

## 2 Methods & Ethics

### 2.1 Methods

The persona of a typical, impact inclined, early- to mid-career UK academic (see Sect. 3) was used to focus and constrain the scope of the work. The overall approach draws upon ideas of reflexivity (e.g. Bostrom et al., 2017) and action research (e.g. Denscombe, 2010; Kemmis et al., 2013); i.e. academics and practitioners considering their work environments, and participating together to solve a problem to produce guidelines for effective practice. Within this for reasons of pragmatism,

time-efficiency and effectiveness, a mixed-methods approach was used. Three sources were used to create the evidential base for this work.



- *Freely available textual data* (i.e. job specifications and promotion/appraisal criteria). These present a university's pre-considered distillation of requirements and aspirations, from multi-level and varied internal and external drivers (e.g. student expectations, national government policy), against which a UK academic will be typically be judged; these may deviate from actual practice.

- The *first-hand experience of a cohort of 17 academics (environmental scientists) and 5 industry-based co-authors*. This is a direct bridge to actual practice, therefore complementary, and un-mediated (i.e. no interposed social scientist). These data provide a view of tasks/criteria filtered through the perception of university-based environmental scientists, and are thus biased. However, the bias is appropriate for this study; perceptions of tasks/criteria are illuminating when forming an understanding of the motives of those doing the perceiving. Whilst

unusual as a form of data collection, working as co-authors is valid and appropriate in this particular instance as this is a familiar, natural and pragmatic mode of engagement for these contributors. Unlike participants in most studies, a pre-existing document is no barrier to offering criticisms and suggesting changes - indeed, quite the opposite is typical. Where doubt existed with respect to comments, 3 semi-structured interviews were used to clarify meanings. Co-authors were selected by two means: (i) on the basis of likely interest in the research from Hillier's network, and

(ii) by volunteers from a list of attendees distributed well in advance of the workshop (see below).

- *A workshop* at NERC's Knowledge Exchange Network (KEN) meeting, 26[th] June 2018 in Glasgow. The session analysed the textual data. 6 participants were from industry. The 21 university-based environmental scientists comprised 7 faculty (i.e. permanent contracts), 11 post-docs and one PhD student, with varying levels of industrial experience. 11 participants are also co-authors.

Details of the methodology are given below with respect to the two questions central to this investigation: What motivates academics to do specific work? And, reciprocally, what might constrain them?

### 2.1.1 Investigating time as a primary constraint

A primary constraint upon collaborations is self-reportedly the time available in an academic's working week (e.g. Abreu et

al. (2009), see above). Thus, a pertinent question is: What do research scientists in universities do? More specifically: What are the duties and responsibilities of a scientist? What are they required to do? Data to answer these (see Sect. 4.2) provide necessary context to understand competing pressures placed upon them as these day-to-day tasks frame what a scientist can do, whatever their underlying desires and motives may or may not be.

Thematic analysis (e.g. Dowling, 2015) was used to build a list of representative, illustrative expectations from the detailed specifications in 10 job adverts (Table 1). Initial review was by the lead author (i.e. Hillier), with Table 1 updated and adapted in light of two rounds of comments from the 17 academic co-authors before the KEN workshop; experience-based context and caveats surrounding Table 1 in Sect. 4.2 are a synthesis of these comments. Finally, at the workshop, participants



rated the statement '*Table 1 is, on balance a fair representation of demands on a UK academic*' using a 5-point Likert scale. Thus, duties in Table 1 are derived to provide a fair level of comparison with expectations within specifications in their number and scope, although some are amalgamated or split here when compared to individual job profiles.

As a crosscheck, participants at the workshop also replicated Hillier's assessment of how often each bullet point in Table 1 was explicitly present in each job specification. The 10 groups of 2-3 participants had one specification, and were instructed to interpret 'explicitly' as they wished.

Arguably, whilst giving a university's considered view on requirements, the tasks in job specifications may deviate from
actual practice. Both the KEN workshop and use of 17 co-authors mitigates this limitation, and allow a view on it to be given (Sect. 4.2).

To obtain 10 job adverts, a non-exhaustive search protocol was used, but one that effectively offers random and objective selection with respect to the information sought; specifically, the adverts used are the first 10 hits from a device located in
the UK for the search 'job description university lecturer' on the Google search engine on 16th May 2018. Only job specifications for advertised posts taken directly from university sites were used (i.e. not agencies or career advice sites). Taking the search results in descending order, 18 were required to find 10 such results. 10 is a relatively small sample of 164 UK universities.  However, after a framework agreement in 2004 involving the University and College Union (UCU), many UK institutions have profiles developed around a set of generic guidance (*https://www.ucu.org.uk/arprofiles*) and so the size
and method of search used here is sufficient to gain a general view.

International specifications were apparently inaccessible without deception (i.e. pretending to apply), so summaries on *jobs.ac.uk* were used; all 11 of appropriate level and type (see Sect. 3) current on 17th May 2018 for Environmental Sciences/Geography/Geology were used (Australia, China, Denmark, South Africa, Sweden, Switzerland, West Indies).
Evaluation of these was limited to broad consistency with the UK profiles and the presence of explicit mentions of impact/enterprise.

**2.1.2 Investigating performance appraisal as a substantive motivator**

How are research scientists in universities assessed in their employment? Despite being a relatively new phenomenon, appraisals (e.g. annual performance review) are now ubiquitous in universities (e.g. Costa and Olivera, 2012; Su and Baird,
2017). Appraisal criteria pertain to strategic aims and aspirations of both university and individual, and are designed to motivate and steer as well as judge (see Sect. 5.2.3), and so differ distinctly from tasks at the day-to-day (i.e. operational) level.





For Sect. 4.3 thematic analysis was used to build a list of representative, illustrative criteria from currently applied, freely available guidance on promotion to Senior Lecturer (see Sect. 3) from 10 UK institutions (Table 2). Initial analysis was by the lead author (i.e. Hillier), with Table 2 reviewed and adapted in light of two rounds of comments from the 17 academic co-authors before the KEN workshop; experience-based context and caveats surrounding Table 2 are a synthesis of these

comments. The co-authors' experience is that appraisal criteria are so strongly aligned with promotion criteria that the two sets of indicators may be treated as synonymous here.

Word clouds were generated to assist understanding; Fig. 1 contains all relevant text from the specifications, whilst Fig. 2 contains words perceived as significant by the KEN workshop participants. Participants considering a set of criteria

highlighted 1-5 snippets of $\leq 5$ words in each of the four main areas (i.e. Research, Teaching, Enterprise/Impact, Administration/Teaching). Whenever categorization in the documentation was different from the main area in Table 2 (e.g. 'Managing People' or 'Pastoral Care'), participants judged which of the 4 areas to identify the contents with.

UK promotion criteria were obtained from all relevant hits of the 190 returned for a search of 'university academic appraisal

criteria' on the Google search engine on $1^{st}$ May 2018 using a device located in the UK. 8 sets of international criteria found concurrently were typically less detailed, and therefore examination was limited to the presence of explicit mentions of impact/enterprise.

### 2.1.3 Pragmatic suggestions for collaboration

The ultimate aim of this work is to suggest *how* an environmental scientist might be pragmatically supported to collaborate

effectively with industry practitioners (e.g. Sect. 5.3). Views here are based on the experience of all 22 co-authors.

### 2.2 Limitations & Biases

In addition to the limitations and biases discussed above (i.e. Sect. 2.1), a number of others exist, but do not invalidate the work.

•   *A UK focus* might be considered to limit the applicability of this study, but assertions of more general applicability are supported by agreement with larger studies (e.g. Sect. 5.1.1) and the co-authors experience; a number have worked as academics in other countries (e.g. Germany, South Africa, USA), and all work closely and openly with international collaborators.

    •   *Bias to the sub-set of environmental scientists participating in this research* i.e. participants and co-authors are

30         inclined towards knowledge exchange (KE). Accepted, but this is the scope of the study (i.e. experience broadly aligns with the persona used).



- *Focus on an illustrative academic persona* (Sect. 3) precludes considering all variants, but sets a basis for future studies.

Many potential avenues are not explored e.g. KE for social scientists where understanding the processes of relationship
building and better collaboration can be 'core business'. Equally, a guide for academics to their industry partner is out of scope, although it may form a future companion paper.

**2.3 Ethics**

Data collected at the KEN workshop was undertaken in accordance with good practice, and clearance was given by Loughborough University's departmental ethics co-ordinator. Contributors to the manuscript were under no obligation to
become co-authors.

**3. Environmental Scientist Persona**

To focus this work and give specific insights, a persona was selected to represent an illustrative university-based environmental scientist. It is a type of person who, whilst not yet deeply engaged in knowledge exchange, a risk practitioner is perhaps most likely to encounter and want to initiate and nurture a long-term relationship with.

Critically, this hypothetical individual's core research and scientific identity involves improving understanding of physical processes (e.g. physics, atmospheric science, geology, hydrology). As such, they can likely make a genuine contribution to risk assessment models for natural hazards. Also importantly, this scientist is assumed to have a genuine and significant interest in impact (i.e. real-world change; also see Sect. 5.2.2), working directly with industry, and has at least some work
that is of interest to insurance or reinsurance in natural hazards. Their level of experience of KE could vary, depending on background (e.g. KE Fellowship, grants, worked in industry), and they may or may not have done reading (e.g. Reed, 2018) or training in impact. Thus, whilst they are primarily judged on scientific research, and KE cannot be their core business (see Sect. 4), it is assumed that any barrier here is in factors (e.g. KE skills, time) other than willingness to try.

They have ≲10 years experience at faculty level. This is regarded as early- to mid- career (e.g. see *https://www.egu.eu/ecs/*), and contains the transition from Lecturer level (L) to Senior Lecturer (SL) level in the UK system; permanent positions (i.e. not fixed-term contracts) typically start at 'Lecturer', then progress through 'Senior Lecturer' and 'Reader' to 'Professor' (e.g. Broch et al., 2017; Wikipedia, 2018a). Faculty level in the North American system has only three stages: Assistant Professor, Associate Professor, Professor (Wikipedia, 2018b). So, a roughly equivalent transition is to Associate Professor.
This stage is ideal to have established a research track record yet still be flexible, and be actively seeking to initiate new long-term relationships.



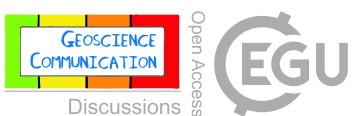

Our research scientist is assumed to be effective and efficient in their work since; quite simply, they would not remain in their position if they were not. This person works the 45-50 hour per week that is *de facto* expected of such an academic (see Sects. 4.1 & 5.1.1, (Bothwell, 2018)); many of those promoted to Senior Lecturer will work more hours, a few less. In this, it

is also assumed that this person has a desire for both a successful career and at least some work-life balance; 50 hours can be achieved in 5 respectable length (i.e. 10 h) working days, or shorter but inclusive of some time at weekends.

At work, in line with the great majority of academics, this person is motivated by career (i.e. funding/reputation) but ultimately by the 'puzzle' (i.e. knowledge/curiosity) (Lam, 2011; Stephan and Levin, 1992) (see Sect. 1). If they were more

interested in the 'gold', they may start a spin-out company rather than engage in the type of joint-collaboration considered here, for example.

A typical university-based job including both research and teaching is also assumed, so our scientist is time-limited (see Sects. 4 & 5). Fellowships won in open competition (e.g. NERC, Royal Society) allow a scientist to focus on a stated work

programme (e.g. in research or knowledge exchange), but are prestigious owing to their relative scarcity. Government funded Research Centres exist (e.g. British Geological Survey in the UK), but staff are required to do applied and income generating work alongside fundamental research. In some countries there are institutions intended purely to do research (e.g. GFZ in Germany), but this is not so in the UK.

Finally, it is assumed that this person is in good mental health, although this is a serious issue in the sector with a notable self-reported negative impact of work for over half of UK academics (Bothwell, 2018), and is a 'rational actor' that in the main responds logically to their internal and external motivations and drivers (e.g. performance appraisals) (Cadez et al., 2017; Grendon, 2008; Harland and Wald, 2018; Moya et al., 2015).

## 4 Results

**4.1 Hours worked**

Faculty level participants at the KEN workshop self-reportedly work a mean of 47.9 hours per week, ranging from 38-70, normalized to 1 full-time equivalent (FTE) where part time. They worked during 18.3 weekends per year on average, with a range from 3-40. These data for UK university-based environmental scientists with an inclination for KE are broadly consistent with the experience and practice of the 17 academic co-authors.

The result gives a view on the spare capacity within an academics' typical week, and so it is pertinent when considering time pressure as a constraint on collaboration (Sect. 5.1.1).



### 4.2 Duties of research scientists in universities

Table 1 illustrates the main duties expected of a typical UK university-based early- to mid-career environmental scientist, namely of ≲ 10 years faculty experience.   There are 22 tasks based on thematic analysis (see Sect. 2.1), roughly commensurate with the median of 28 'key' or 'main' duties and responsibilities in the job specifications analysed; the range is

15-52 tasks. The consensus of the 17 academic co-authors is that this, including the time allocation, broadly reflects our experience of UK universities. Similarly, in the KEN workshop more than twice as many participants agreed as disagreed (12 vs. 5) with the statement that '*Table 1 is, on balance a fair representation of demands on a UK academic*'. Notable details from the table and experience-based caveats of the academic co-authors to it are reported below, including extra detail from the original job specifications where it is useful.

**[ TABLE 1 HERE ]**

Teaching will readily expand beyond 2-days a week on average if permitted to by the researcher, as will administrative duties, and this load is spread unevenly throughout the year; it is common for little research (including impact) to be possible

in term times, with a real possibility that none is possible for 1-2 months during a busy term (i.e. if an imbalance in teaching load between terms exists). This effect becomes severe if programmes or modules need to be rewritten or restructured, which can take hundreds of hours whilst other demands do not lessen.

Duties occurring in most job specifications (black type in Table 1) are all time-consuming requirements. However, in the

experience of the academic co-authors those with low numbers of occurrences are also ubiquitous (e.g. reviewing funding bids and papers written by others) and illustrate the numerous other activities a researcher is simply expected to find time for (e.g. see Lazarsfeld-Jensen and Morgan, 2009). Several activities to show leadership outside the university are also usually required. Examples of such roles include journal editing, sitting on panels assessing funding bids, roles such as treasurer for learned societies (e.g. British Society for Geomorphology), invitations to be an examiner at other universities, sitting on

government committees, working with funding bodies to define future research directions, and outreach (e.g. Pint of Science *https://pintofscience.co.uk/*). Note that a scientist's own, hands-on research activity (i.e. doing it rather than managing it) forms a relatively small part of the 2-days per week allocated to 'Research', and impact (a.k.a. innovation, consultancy or knowledge exchange) is only a sub-part of this.

Professors and Readers are usually also expected to undertake more substantive management roles (e.g. Head of Department, lead of a Doctoral Training Centre, Admissions Tutor, Programme Director), and other requirements are sometimes reduced to account for such a time commitment.



Job specifications associated with adverts outside the UK are typically less detailed. However, a brief review of 11 specifications (Australia, China, Denmark, South Africa, Sweden, Switzerland, West Indies) indicates that roles and responsibilities worldwide are broadly similar to the UK. Out of the 11 specifications, 5 explicitly mention impact or similar (cf. Table 1).

In summary, university-based environmental scientists typically work in a highly time-limited environment and have an array of competing demands upon their time. If efficient and effective in their approach, experience suggests that about 1 day per week can be protected in which to do a scientist's own hands-on research (funded or otherwise), of which perhaps *up to* half a day can be committed to impact (e.g. insurance focused implementation).

These day-to-day tasks frame what a scientist can do, whatever their underlying desires and motives may or may not be. Thus, these results provide necessary context to understand what specific work our illustrative scientist will or will not be able to do, and why (Sect. 5.1.1).

**4.3 Criteria used to assess research scientists in universities**

15 Table *2* is an indicative set of criteria for an early- to mid-career UK academic, based on thematic analysis of promotion criteria to Senior Lecturer (Sect. 2.1). On balance, and taken as an illustrative realisation of a more complex totality, these are a fair representation of assessment criteria in the experience of the 17 academic co-authors.

Of the four main areas (i.e. Research - R, Teaching - T, Enterprise/Impact - E/I, Leadership/Administration - L/A), all but E/I 20 are always present as a main heading within the criteria (Table 2). E/I is a main heading in only 3 of the 10 institutions (30%), although detailed examination of the documents reveals that criteria relating to E/I are pervasively present in all UK institutions. This is consistent with the knowledge and experience of the 17 academic co-authors, and contrasts with international universities where only 2 of 8 had 'Public Service' & 'Knowledge Transfer' prominently featured and in another 5 E/I was entirely absent.

25 **[ TABLE 2 HERE ]**

To be viewed as acceptably meeting expectations, good performance in at least 2 of the 3 traditional categories (i.e. R, T, L/A) is typically required in the UK; this is based on co-authors' experience and examination of the criteria documents. Word clouds below directly display an impression of key aims from the underlying text (Fig. 1), and how the aims were perceived 30 by academics at the KEN workshop (Fig. 2). Notable elements of the word clouds are summarized in the sections below (Sects. 4.3.1-4.3.4), accompanied by explanation based on the co-authors' experience and reading of the underlying texts; a summary of the aspirational criteria used to assess UK academics precedes this.





In short, publishing novel science in peer-reviewed journals is the overriding imperative, then winning funding to facilitate publications (i.e. by funding a post-doctoral researcher). Teaching and Administration/Leadership are obligatory. Pervasive pressure (i.e. criteria) exists to undertake Impact/Enterprise work, in whichever diverse form, but in practice it remains lower in priority, is not obligatory, and is best engaged in if reportable outcomes also aligned with other drivers. To the 17 co-

authors knowledge, it is similar outside the UK, except that institutional pressure for impact is usually lower; however, such pressures are developing (e.g. in Australia, Germany).

Thus, these results give an indicator of how our illustrative scientist may respond (e.g. in terms of prioritization) to time pressure within the work context (Sects. 5.1.1, 5.2.3) and direction from funding bodies (Sect. 5.1.2).

### 4.3.1 Research

Key words for promotion to Senior Lecturer show the need for a sustained high-quality research record (i.e. publications), and funding (Fig. 1a). Academics' perceptions focus on these even more dramatically (Fig. 2a). These are again repeated in our illustrative, representative profile (Table *2*), but this also includes an emphasis on PhD supervision, and reputation. These may appear disparate, but in the experience of the academic co-authors are strongly bound together.

A university scientist's international reputation is built almost entirely on novel, high quality, well cited peer-reviewed publications (i.e. journal papers); these evidence a research profile and incomplete (i.e. unpublished) work is of little value. Funding must be underpinned by related publications, with some flexibility to take moderate steps in new directions driven by curiosity, and provides the resources (e.g. post-docs) to create excellent publications. As well as developing the next

generation, PhD students can be an effective means to generate publications in a time-limited university environment.

### 4.3.2 Teaching

Key words for promotions to Senior Lecturer show the need for significant student-focussed teaching of quality (Fig. 1b). Academics' perceptions also include an emphasis on development, design or innovation, and a 'HEA-Fellowship' is present

(Fig. 2b). The illustrative, representative profile (Table 2) adds context, such as for the 'HEA-Fellowship'; in the UK a professional qualification with a body such as the Higher Education Academy (HEA) is required to evidence attainment in teaching. Also, Table 2 explains 'develop'; this could be of new module (e.g. a set of 10 lectures and practical sessions) or programmes (e.g. a new 'Global Environmental Risk' BSc), although there is an expectation of innovative and stimulating modes of delivery (e.g. experiential, problem-based learning, integrating iPads/tablets). In the experience of the academic co-

authors, student satisfaction is very important in practice as measured internally by module or programme feedback and externally by the National Student Survey (NSS, *https://www.thestudentsurvey.com/).*





'Research-led' teaching based upon a scientist's core research is required, but the teaching does not feed back into the scientific research. However, teaching is typically the main source of university funding (Universities UK, 2016), and a key reason universities exist, and monitored in the national Teaching Excellence Framework (TEF) assessment (*http://www.hefce.ac.uk/lt/tef/*), so this is not typically optional (see Sect. 4.2). The expectation is to design, maintain and

deliver customized material that exceeds that to be found in textbooks, increasingly based on the academics own published interests as a degree proceeds.

### 4.3.3 Enterprise / Impact

Enterprise occurs frequently as a key word within promotion criteria for Senior Lecturer (Fig. 1c), but neither academics' perceptions from the KEN workshop (Fig. 2c) or our illustrative, representative profile (Table 2) are able to focus on

specifics.  Examination of the underlying criteria and words used to generate the clouds show that this is due to the range of possible activities here.

The academic co-authors' experience indicates that, whilst of increasing importance, Enterprise or Impact activity is only considered of value if it generates income to fund future research or is suitable for a REF Impact Case Study, and ideally

facilitates or inspires better curiosity-led research (see Sect. 5.2.2).  This said, pressure to engage in impact-related work is pervasive in the UK from institutions and funders (see also Sect. 5.2.2).

### 4.3.4 Leadership / Administration

Key words for promotion to Senior Lecturer show the need for evidence of contributions to the Department/School and University (Fig. 1d), whilst academics' perceptions highlighting that this includes leadership and leading externally (e.g.

driving national and international initiatives, promoting a university's brand) (Fig. 2d). The descriptors in our illustrative, representative profile (Table 2) are more explicit stressing leadership, success and innovation. Outreach (e.g. public talks, 6[th] form summer research experience) is encouraged, but is essentially optional, and delivery of all standard administrative tasks (e.g. research team management, undergraduate module leadership) is taken as read. L/A will not get an academic short-listed for a job or promotion, but evidence of competence in this it is required for them to actually get it.

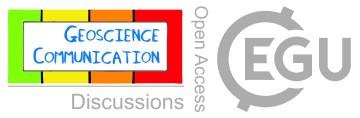

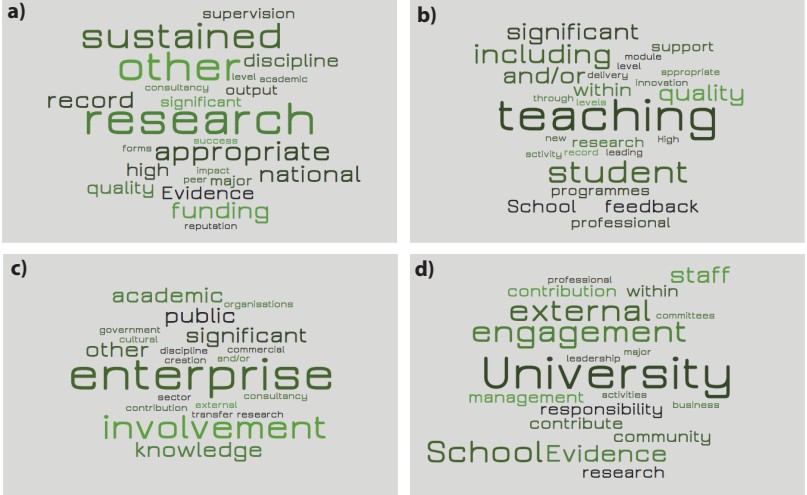

**Fig. 1: Word clouds summarizing appraisal/promotion criteria for 10 UK universities at Senior Lecturer level in the 4 main assessment headings a) Research b) Teaching c) Enterprise/Impact d) Leadership/Administration. Minimum frequencies vary from 2 to 5 to give 20-30 words displayed. Sizes according to rank.**

**Fig. 2: Word clouds summarizing the KEN workshop participants' perceptions of promotion criteria for 10 UK universities at Senior Lecturer level in the 4 main assessment headings a) Research b) Teaching c) Enterprise/Impact d) Leadership/Administration. Minimum frequencies of 2. Sizes according to frequency.**

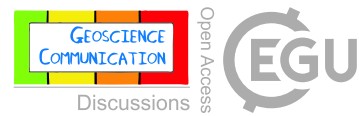

## 5 Discussion

Successful university-business collaboration requires mutual understanding, built upon shared vision and long-term trusting personal relationships (e.g. Dowling, 2015). In building collaborations industry practitioners will be assisted by understanding the answer to two questions: What motivates academics to do specific work? And, reciprocally, what might
constrain them? This discussion provides a window into the motives of university-based research scientists that, in addition to practitioners, will be highly relevant to a number of academic colleagues, university administrators and policy makers.

Constraints on collaborations are considered first, in Sect. 5.1. Then, environmental scientists' motivations are discussed in Sect. 5.2, culminating in a conceptual model of academics' motivations (Fig. 3). Section 5.3 considers the practical aspects
of building an industry-university collaboration based on a 1-to-1 relationship; an illustrative, non-exhaustive list is proposed of pragmatic suggestions for short-term and time-efficient activities that have reportable and mutually-beneficial outputs for both an academic and a risk practitioner in order to build the long-term trusting relationship needed for collaboration.

### 5.1 What constrains scientists' working with industry?

65-89 % of university scientists, depending upon country, have a prime interest in research rather than teaching (Abreu et al.,
2009; Cavalli and Moscati, 2010). However, 'Teaching' is not optional (Sect. 4.3). This leads to a conflict for limited time between teaching and research (Sect. 4.3), which is well known (Arnold, 2008; Gendron, 2008; Harland and Wald, 2018; Moya et al., 2015), even if the consequences of this continue to be debated - see the summary in Cadez et al. (2017). For an environmental scientist, work with industry is based upon their science and this study shows that for workload purposes this typically falls within time allocated to research (Table 1). This section expands the debate by incorporating impact into this
tensioned research-teaching relationship. Specifically, it considers the influence of time pressure due to workload factors (Sect. 5.1.1), the role of funders (Sect. 5.1.2), intellectual property (Sect. 5.1.3), and academics' need for a coherent track record (Sect. 5.1.4) as potential constraints upon university-business collaboration.

### 5.1.1 Time pressure

A self-reported mean of 47.9 hours/week is worked by the sample of UK-based environmental scientists involved in
knowledge exchange (Sect. 4.1). This is consistent with larger studies. The Changing Academic Profession (CAP) survey of 100 academics (2004 to 2012) (Teichler et al., 2013) describes a self-reported mean workload of ~48 h/w across 18 countries; 45-50 h/w in the UK is in line with Australia and about mid-spectrum (Fig. 17 of Coates et al., 2009), and is supported by recent data from 2,000 academics (Bothwell, 2018). This time at work sets the boundary conditions for accomplishing the 15 to 52 distinct 'key' or 'main' tasks required of a university-based scientist, in addition to which there is
an expectation to do numerous other tasks to support their academic reputation, internal visibility and external profile (Sect. 4.2). Thus, even working ~50 hours per week, it is evident that there is time pressure for a typical university-based scientist


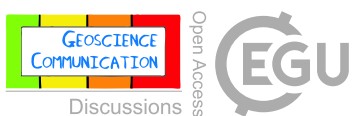

in the UK or elsewhere. As Teichler et al (2013: p99) notes, the presence of stringent time constraints is certainly not unique, and applies equally to other high-skilled jobs (e.g. in the insurance sector). So, this work confirms that an academic with days and days to sit, gazing around and pondering, is a myth with no evidential base in today's universities.

Table 1 reports a ratio of approximately 2:2:1 for R:T:L/A in the UK, with the nature of the need to juggle these demands described in Sect. 4.2. Table 1 is entirely consistent with the CAP survey (Teichler et al., 2013), making it widely relevant. CAP reported teaching as 38-46% of work hours, and ratios of research to teaching in a range of countries (e.g. UK, Finland, Portugal) nearly parity. The need to juggle R, T and L/A demands is also reported for Australian institutions (Coates et al., 2009; Lazarsfeld-Jensen and Morgan, 2009). Interestingly, Australian Professors work 52.2 h/w, more than SLs at 46.4 h/w

with the difference made up by research. In other words, with other things (i.e. T, L/A) non-optional, a stronger staff profile appears to be created by working more hours to do research (Coates et al., 2009); e.g. 100% of 91 Australian academics reported working weekends, 43% of these in the 37-48 weekend per year bracket (Lazarsfeld-Jensen and Morgan, 2009). This aligns with the co-authors' view of the UK system although, critically, the introduction of Enterprise/Impact in addition (Tables 1 & 2) and pervasive pressure to act on this (Sect. 4.3) has increased the difficulty of the time-management challenge

amid competing demands.

In addition, Table 1 adds clarity to what is covered by 'Research'. Specifically, numerous other tasks (e.g. PhD supervision, preparing funding bids, grant-related administration) are present that might not immediately be thought of when considering an academic doing research. If research is considered to include reading articles, modelling, programming, learning any new

skills required, writing journal articles, and any additional work with industry, about 1 day/week might be retained for this; this level of detail sheds light into what might constrain scientists' work with industry in the UK and globally.

In short, the critical new observation (Sect. 4.2) is that an efficient and effective academic might retain about 1 day per week in which to do their own 'hands-on' research; of this perhaps *up to* 0.5 days can be committed to industry (e.g. insurance)

focused implementation (i.e. impact). Prioritization is key, and any choice to do something is inevitably a choice to discard an opportunity of less potential value; often, an impact task will be competing directly against the little curiosity-driven research that is possible, time with family or children, or weekend recreational activities (e.g. see Bothwell, 2018). A convincing (self-)justification is likely therefore needed well before any official appraisal.

### 5.1.2 Direction from funding bodies

Ultimately, the topics, scope and even existence of environmental science research are set by funding bodies. Funders may be government (e.g. NERC in the UK, ERC in Europe, NSF in the USA) or industry (e.g. Willis Research Network, AXA Research Fund).  Government funding (e.g. UKRI) may be either specific to particular projects through grant bids to particular funding 'calls', and eligibility may be restricted to certain topic areas, geographic locations or status of applicant.



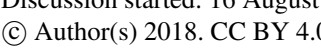

Conversely, if a business is prepared to fund a particular topic then there is a good chance it will get done. It is also notable that UKRI funding now typically incentivises collaboration with industry (see Sect. 5.2.2).

### 5.1.3 Intellectual Property

Dowling et al. (2015: p27) review the barriers to industry-business collaboration (e.g. identifying partners inside insurers
(Abreu et al., 2009)). One notable barrier comes from tight expectations about knowledge ownership and exclusive use (e.g. Dowling, 2015). This can be seen as a fundamental mismatch between creating benefit for society directly by publishing (i.e. not supporting the profits of one company) and supporting the public only indirectly via tax of increased revenue or better insurance products. Ultimately, academics cannot compromise on publication.  Illustratively, a compromise is to use post-project embargo periods for publications or data (e.g. 6-12 month) (Drexl, 2016; e.g. Morris et al., 2011; Moulin, 2018;
UKRI, 2018b), perhaps explaining relatively low levels of concern amongst individual academics about this issue (i.e. Abreu et al., 2009). The insurance sector is a multi-national illustration that these considerations are similar across the globe.

### 5.1.4 Track Record

Globally, international reputation is important both directly in appraisals (e.g. Table 2) and obtaining funding and requires a track record in specific activities (Sect. 4.3.1). Thus, there is some need for continuity, which may be perceived as 'pet
projects' by industry.

### 5.2 What motivates research scientists to do specific work?

The majority of new knowledge that could be used by industry (e.g. insurance - see Sect. 1) is published by university-based scientists in journals. How can practitioners best access and harness this existing knowledge, and work with these researchers to answer new questions as they arise? To contribute to answering this question, this section considers
academics' motivations in light of the results of this study, explicitly translating generalities of academic debates into the context of environmental science using illustrations from insurance. With basic needs met (see Olsen, 2004), additional personal financial reward (i.e. 'gold') is of low importance to the great majority of researchers (e.g. Abreu et al., 2009; D'Este and Perkmann, 2011; Evans, 2016; Lam, 2011), who do little or no consultancy work; so, for a risk practitioner, it doesn't matter how much you might be able to pay them to work with you.  Persuading the world's best researchers to work
with you requires a deeper understanding of what motivates most academic researchers.

The findings of this study (Sect. 4), and experience of the co-authors, leads us to propose impact as a notable addition to prior models (e.g. Lam, 2011) (see Sect. 1).  Broadly speaking therefore, after 'gold' there remain three types of inter-linked motivation (i.e. Fig. 3) influencing our illustrative research scientist. Each of these presents an opportunity for a risk
practitioner (see also Sect. 5.3).





1. *Curiosity and creativity* (a.k.a 'puzzle' (Lam, 2011)): By temperament, given unlimited time and funding, academics would simply study whatever interests them most for the satisfaction of a puzzle solved in an innovative way. How can you frame your needs in a way that will pique the curiosity of researchers, challenge them and give them opportunities to conduct creative, original and publishable work?

2. *Impact*: *S*ome academic researchers want to make a positive impact upon society (Reed, 2018) (i.e. 'altruism'), whilst others are intrinsically motivated by the act of working with industry itself (i.e. 'utility'). How will working with your company give these researchers a unique opportunity to make a difference that is significant and meaningful, and at a scale they could otherwise only dream of?

3. *Career* (a.k.a 'ribbon' (Lam, 2011)): Increasingly, generating such benefits in the real world is now rewarded, with
some contribution to winning research funding and promotion (see Sect. 4.3). How can you provide evidence of impact from research that can be used by researchers in evaluation exercises?

These drivers are considered below.

### 5.2.1 Curiosity and creativity

Curiosity is a major driver for most researchers (e.g. Lam, 2011), who want to be at the cutting edge of their discipline. The excitement of discovering something new can be addictive, even when the breakthrough seems elusive, and many researchers are motivated by the intellectual endeavour required to overcome the challenges that stand in their way. Sometimes the journey is as rewarding as the destination, as researchers are forced to engage with new disciplines and ways of thinking in their pursuit of creative solutions. Also, the challenge of coming up with new solutions to old problems should
not be forgotten.

The results of this study (Sect. 4) in no way contradict existing views of creativity and curiosity. Fundamentally, curiosity is the seed from which all academic publications grow, and publications remain central to international academic reputation and appraisal (Sect. 5.2.3). However, the results reveal the bounds (e.g. time around other duties) in which curiosity must
operate. This effectively limits the utility of vague, unconstrained or highly-speculative curiosity; such tasks are unlikely to rise to the top of a list of pending actions. It is therefore important to focus and formulate questions that are precise enough for the scientist to be *able* to answer, and intriguing and novel enough for scientists to *want* to prioritise answering them.

So, how can a practitioner (e.g. in insurance) tap into this set of motives?

- Rather than simplifying the nature of the challenges you need to address, can you explore the complexity of the challenge, and ask 'why' questions that cultivate your own sense of curiosity in the challenge as something to be understood, not just solved.


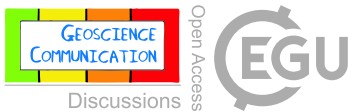

- Before engaging with academic researchers, have you checked that there isn't already an answer to your question in the research literature? Google Scholar has made it easier than ever before to access published research. Use what you learn from your reading to put your question into the context of what is already known, and explicitly articulate what is not yet known; this is an ideal way of both identifying an academic researcher and framing your approach to them. Alternatively, you might use such scoping exercises as a mechanism for collaboration building (see Sect. 5.3)

- Consider what unique opportunities you can give to a researcher who loves the creativity of what they do. Can you expose them to new ways of working or thinking, introduce them to colleagues who ask challenging questions or expose them to methods used in the business world to drive original thinking and innovation?

- Actively promote (e.g. host events, provide needs-based rationale to pursue) multi-sector collaboration, which opens new avenues for innovative research (e.g. across traditional subject boundaries)

### 5.2.2 Impact

'Impact' is a term used to describe the influence that underlying research has outside academia (Reed, 2018). In the UK government bodies, i.e. NERC and HEFCE (HEFCE, 2015) now merged into UKRI (*https://www.ukri.org/*), define impact broadly as

*'An effect on, change or benefit to the economy, society, culture public policy or services, health, the environment or quality of life'*

*S*ome academic researchers, especially in applied disciplines such as environmental science, have trained because they intrinsically want to make a positive impact upon society (Reed, 2018); i.e. 'altruism'. Alternatively, we assert that others, including a group of the co-authors, are motivated by the act of working with industry itself, assisting pragmatic implementation and being useful in that way; illustratively hearing '*we can use this*' or '*that'd be really valuable*' energises these academics. This may be dubbed 'utility'. Whatever a scientist's exact internal motivations, however, the findings of this study (Sect. 4), highlight that impact work must align with other demands of time such as research and teaching that are currently considered more important for the role of an academic and for promotion; this is despite the recognition of impact in job descriptions and promotion criteria.

This research also demonstrates that for a research scientist's job, it is critical to be able to evidence impact, demonstrating benefit from their research (e.g. behaviour change, competitive advantage in business, attracting foreign investment, new or changed policy); without evidence it is effectively useless to them for appraisals. In the UK impact is being driven into the appraisal structure by the government funding councils' inclusion of Impact Case Studies in their assessment of research excellence (REF), and whilst an administrative burden this could be a key mechanism to encourage effective collaboration.


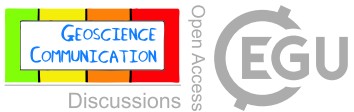

All funding proposals to the UK Research Councils require the creation of an impact summary describing who will benefit from the research and pathways to impact describing the approach that will be taken to deliver these impacts. Although traditionally weighted significantly lower than scientific excellence, one recent large funding scheme (the £4.7 bn Industrial Strategy Challenge Fund) weights impact only slightly less than excellence, and in another (the £1.5 bn Global Challenges

Research Fund) it is the main objective (UKRI, 2017, 2018a); thus, applications to the funds require credible, significant and far-reaching impact for proposals to be fundable. The importance of impact is also growing in the UK's 7 yearly appraisal of research across the Higher Education sector, now accounting for 25% of institutions' scores and significantly affecting league table rankings and income (i.e. REF2021 http://www.ref.ac.uk/). Internationally, Australia's Excellence in Research for Australia (ERA) exercise in 2018 (i.e. http://www.arc.gov.au/era-2018) is now partnered by an Engagement and Impact

Assessment (EI, *http://www.arc.gov.au/engagement-and-impact-assessment*), highlighting the growing weight given to impact in a number of countries.

Within the insurance sector, we propose that types of impact (Reed, 2018: Ch 2) and supporting evidence might include the following, although which is most important varies by a practitioner's role (e.g. broker, research manager, model developer):

1.  *Cost savings* (e.g. saving on re-insurance), or *increased profit* or an increase in turnover (e.g. by better pricing) where the research made a significant contribution to decision-making that led to the benefit. Economic benefits such as these may be evidenced via

    a.  financial records (these can be clearly marked for the eyes of reviewers only and redacted for any public

record)

    b.  reports in mainstream media or industry publications (e.g. Insurance Times), ideally stating the change or difference that has been made and linking this to the research

    c.  a testimonial letter describing the nature of the benefit in the risk practitioner's own words, and how it arose from work with the researcher

2.  *Improved strategic decision-making* e.g. entering a new partnership or geographic region based on evidence from the research. Decision-making impacts like these may be captured in strategic documents and agreements; citing the published research in these documents makes it easier for researchers to claim impact. Otherwise, testimonials are widely used to evidence this sort of impact

3.  *Capacity-building impacts such as new skills* or business capabilities generated via internal training courses by

researchers based (at least partly) on their research; evidence for this could include the amount of training conducted, feedback from participants ideally indicating the effect the training has on their work, or any publicly available white paper, policy document, professional newsletter or blog stating the advantage gained through the research




4. *Understanding or awareness impacts* such as uncovering the scale or urgency of a problem, perhaps of a peril (e.g. clustering of extra-tropical cyclones (e.g. Vitolo et al., 2009)). There may be no solution to the problem at present, such as for flood-wind interdependency (e.g. Hillier et al., 2015; De Luca et al., 2017), but awareness may have an impact in itself, and in time lead to further impacts. Evidence could relate to recognition in blogs or industry awards

(e.g. Lloyds' Science of Risk Prize).

**5.2.3 Career**

Academic systems vary by country (Cavalli and Moscati, 2010; Coates et al., 2009). However, in general, tenure with its guaranteed job security (Adams, 2006) has declined (e.g. in the USA), or been eliminated entirely (e.g. UK) (Finkelstein, 2010; Huisman et al., 2002). In Germany a job-for-life remains, but since the 1980s in the UK and Netherlands, university

staff are employed by their institution and not the state, and in 1988 the UK government legally abolished tenure (Enders, 2015; Legislation, 1988). This opens up university research scientists to a much greater steer by appraisals (e.g. Costa and Olivera, 2012; Su and Baird, 2017) and via promotion criteria that are the universities' distillations of institutional and external policy expectations (Sect. 4.3).

A main finding here (i.e. Sect. 4) is that national level policies to incentivise impact (i.e. REF and funding, Sect. 5.2.2) have now entered into the everyday consciousness of UK academics, with pervasive pressure to engage in impact-related work from institutions and funders; whilst Enterprise and Impact have propagated to be main headings in only 3 of the 10 institutions considered, all promotion documents contain criteria relating to E/I. In response, many academics now pursue impact to align with institutional requirements. Notably however, in terms of time-allocation and duties, impact-related work

is one task amongst many (i.e. Table 1) and is likely only considered of value if it generates income to fund future research or is suitable for a REF Impact Case Study. Evidentially, in practice, it also remains subservient in importance to research and teaching, thus it is wise and perhaps critical for work with industry to facilitate or inspire better curiosity-led research (see Sect. 5.2.1). Lastly, to complete the contextual picture, it is necessary to understand that only a minority of UK academics are required to be heavily involved in KE (e.g. Reed, 2018); with ~1 REF case study per 10 academics, required

involvement is roughly 20-30% of researchers.

Based upon the data in Sect. 4, Fig. 3 presents a simplified model of the task facing an early- to mid-career university-based environmental scientist on the teaching and research pathway most commonly available, modified to include work with industry; it integrates international (i.e. Netherlands, New Zealand, North America, Slovenia, Spain, UK) literatures on

motivation (D'Este and Perkmann, 2011; Freitas and Verspagen, 2017; Lam, 2011) and evaluation (Cadez et al., 2017; Grendon, 2008; Harland and Wald, 2018; Moya et al., 2015), building upon a view based around a teaching-research dipole and older (i.e. pre-impact emphasis in UK) ideas (Hughes et al., 2008). The model assimilates some international data (Sect. 4.2) and the 17 academic co-authors' experience, both first-hand and from cumulative decades of informal discussions of

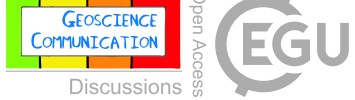

practice with international academic colleagues (e.g. at conferences). Thus, we tentatively posit that it is widely applicable, or is at least a suitable basis for future discussion; a key variant will be the strength of the impact-career link.

Publishing novel science in peer-reviewed journals is an academic's overriding imperative (e.g. Hattie and Marsh, 1996),
followed by winning funding to facilitate publications (i.e. buy a post-doc's time). Publications (bold box and arrows on Fig. 3) are the critical appraisal measure as they underpin teaching, impact, career (i.e. promotion, mobility, or simply retaining a job), reputation and future funding bids (Sect. 5.1.4). 'Funding', or more generally resources (i.e. PhD student time, post-docs, £), is an important appraisal criterion as it indicates reputation and ability to do research, but is significant to an academic as a career measure in its own right. Research itself is marked lightly (Fig. 3) as it is ascribed little value until
published. Impact is light as its influence upon an academic's assessment is still relatively limited, although research and impact pathways are now possible at some universities.

The entrepreneurial route directly between research and impact (Fig. 3) is indicated with a dashed line as it is relatively uncommon (Lam, 2011), and the arrow from impact to funding is thin to reflect the current relative influence it has on the
magnitude of resources.  Teaching and Administration/Leadership are obligatory, but will not get a scientist short-listed for a job so are not focussed on in Fig. 3, even if they are required to actually obtain the job.

The important thing for a risk practitioner to recognise is that if an academic already working ~50 h/w (Sect. 5.1.1) wishes to engage with industry, their only solution is to be effective and efficient, and prioritize carefully to select what they will not
do; and, usually the only moving part is their own research.  Thus, such real-world impact must inspire curiosity and provide some way of better doing new science (e.g. ideas, access to novel data, resources, a PhD student), at least in the longer-term; see Stokes' (1997) dynamical model including 'use inspired basic research' for theoretical context (e.g. Cantisani, 2006: Figs. 2 & 3). If this feedback exists, the academic may be able to find up to about half a day per week, but such a large time sacrifice would need substantial incentives.




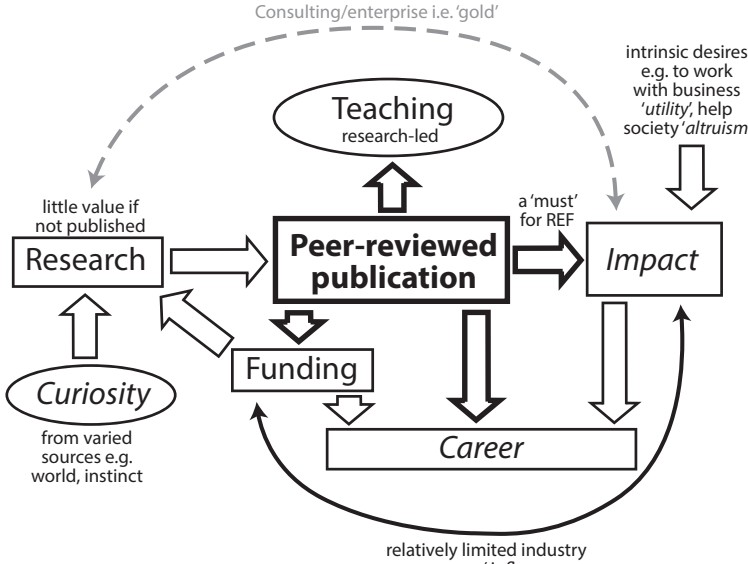

**Fig. 3: Summary of relationships between academics' motivations. Details (e.g. bold or dashed lines) are explained in the main text, and 3 main motivations after 'gold' discounted are in italics.**

5   **5.3 Practical hints and tips to build collaboration**

The objective of collaboration is to translate business-relevant questions (Dixon et al., 2017; e.g. Lighthill Risk Network, 2016) into research questions that are precise enough for scientists to be *able* to answer and intriguing and novel enough for scientists to *want* to prioritise answering them, then deliver outputs of benefit to all. A vital element of successful collaboration is a long-term trusting relationship (e.g. Dowling, 2015) which, despite being an on-going and time intensive

10   venture to construct, is critical in overcoming a number of barriers such as IP (Sect. 5.1.3), language (Fish and Saratsi, 2015; e.g. Jordan and Huitema, 2014; Scott et al., 2018)), differing working and problem solving cultures (e.g. Amabile et al., 2001), or simply contextual knowledge and credibility (e.g. Hughes et al., 2008); just as an academic's motivations and skills may not be transparent to the wider world, academics do not instinctively have an intimate knowledge of any insurer's technical approach, recent initiative, or strategic internal drivers. Of particular relevance is the difficultly that novelty (i.e. for

15   publications) and immediate industrial implementation (e.g. in operational risk models) are not directly compatible in the same task, although steps that will mainly benefit one party or both can be balanced over time throughout a co-designed project.



Many conceptual models of the knowledge exchange process exist (e.g. see Jacobson et al., 2005; Ward et al., 2009). No particular one is assumed, although cyclic, iterative and two-way elements are recommended.

Below we propose two illustrative, non-exhaustive lists of pragmatic suggestions for time-efficient activities that have reportable and mutually-beneficial outputs in order to build the long-term trusting relationship needed for collaboration between an academic and a risk practitioner. Each suggestion includes an explanation of *why* the activity has benefit/utility to justify time spent on it, mapped back to appraisal criteria (Table 2) or an impact typology (Sect. 5.2.2) (Reed, 2018). Obvious relationship-building and maintenance activity is assumed; e.g. short chats over a coffee, telephone calls, passing on interesting items (e.g. a paper or newspaper clipping), and mutual tolerance of unavoidable busy periods (e.g. Sect. 4.2).

The following list details ways in which risk practitioners can support an academic partner, including a brief commentary on how and why benefits emerge. Square brackets e.g. [2] indicate mapping to criteria by which academics are assessed in Table *2*. Suggestions are not ordered as their relative utility will be case specific.

• *Write letter of support/engagement on a research grant* [4]. A useful relationship building measure, and will be best when projects are co-designed. They are required for UK funding applications, but if sought at the last minute after limited discussion it often remains unclear why the insurer should priortise this action (i.e. what the benefits of the work might be), potentially leading to mutual frustration. In contrast, a timely discussion leading to a letter of support which indicates the scale of potential impact of the research to the industrial partner, and details pathways
to it (e.g. specific committees, regulatory compliance requirements, or internal initiatives), would significantly strengthen a research grant application.

   • *Offer a place on an advisory panel, executive committee, or similar* [8,9 and potentially 3,4]. Even with a non-disclosure agreement (NDA) in place, this could be useful to an academic on many fronts; a role title enhances their CV/annual appraisal, even a small remuneration (e.g. ~£1,000 across a year) looks good as an income source from
industry. Furthermore, as such monies are undesignated they are ideal to buy a little of a research assistant's time (e.g. a PhD student) to do a pilot study; these greatly help when writing funding bids for substantive money, ultimately leading to publications. If the academic were also able to say the advice stemmed from a published paper, this would be evidence of impact outside academia, particularly with a supporting statement (e.g. '*advice on earthquake clustering was provided, drawing on X's recent publications, contributing to an evaluation of two*
*catastrophe models*'). On both sides, it is another chance to meet, talk, and build a relationship.

   • *Request a few (e.g. 1-3) days consultancy* - e.g. commenting upon a catastrophe model's documentation [8,9 and potentially 3,4]. The money and impact benefits are as above. This is likely a loss leader for the academic, but is something useful that is safe, will happen, and is measurable and reportable. In some ways this is the mirror of the literature review that they can do for you when written into a project grant (see list below).




- *Provide access to data* - [9 and first step to 3,4] to allow a novel insight into a scientific problem; this would be very useful for the academic and pique their curiosity, but will likely need significant funding to fully use. This said, a data-driven pilot study could be of immediate use to the risk practitioner and also give a strong core to an academic's bid for funding.

- *Offer grant funding for 'innovation' or highly applied work*, either directly (e.g. AXA's research fund) or via government funding that requires an industrial lead (e.g. Industrial Strategy Challenge Fund, Innovate UK) [4,8,9]. This is good for an appraisal's funding metric, and can provide a means of moving towards evidence of impact outside academia, but it is unlikely to produce highly-novel, inventive, cutting-edge scientific research. So this is useful either in the relatively short-term (1-3 years) or in parallel with blue-skies funding, but is insufficient alone in the longer-term for the scientist.

- *Fund blue-skies research* [3,4]. Brilliant! But, the sums are relatively large as with full economic costing in the UK a post-doctoral research assistant for the 2-3 years needed to make a post attractive to a good candidate costs ~£200,000. A PhD student (e.g. via CASE awards) could be much cheaper, but mechanisms to control what exactly they research are limited once they start, they will take longer as they need training, and a chance of failure must be considered; as such this route has added risk.

- *Collect evidence of impact* [9]; impact is diverse, and evidence not onerous to obtain (see Sect. 5.2.2). Creating an Impact Case Study for the REF exercise can win internal (i.e. university) investment in the form of time or money, freeing the academic to pursue further research or develop this strand of impact.

- *Co-design a research project* [3,4,9]. Impact designed in at a project's inception (e.g. Reed, 2018) can inspire world leading science and publications in the highest-impact journals, with ideas and inspiration possible from all parties. However, preparatory conversations over time are needed to ensure there will be novel insights into the underpinning physical processes at work as well as real-world impacts. Early in a collaboration this may take substantial time (e.g. 6-12 months). It is important to note that the impact and novel science do not need to come at exactly the same time or from precisely the same task; i.e. distinct outcomes particular to both risk practitioner (see below) and academic (e.g. publications) should be separately identified. Ultimately, even if both scientist and practitioner are time-limited, co-design in an established relationship can be efficient i.e. a route to better research, faster (e.g. Amabile et al., 2001).

- *Ask them to provide training* - If a clear fit exists, paying an academic to provide in-house training is a good way to get to know them, which the academic can justify in the same way as consultancy.

- *Provide access to training, expertise (e.g. actuaries) or networks* - primarily a mechanism to maintain contact and alignment, since academics are typically proficient at obtaining these already.




Although apparently a counterpoint to the main theme of this article, aimed at risk practitioners, an illustrative list of actions a university scientist may take to support their risk practitioner is given below; it may assist practitioners new to the role of collaboration with academics or as an aid to give to an academic new to collaborating with insurers.

In this spirit, it is worth giving a precis of motivations within this industrial sector. As individuals, it is notable that practitioner's motives are mixed, with curiosity (i.e. the 'puzzle') and family common drivers, not just the 'gold' (see Sect. 5.2, (Lam, 2011)). Whilst insurers ultimately require increased profitability, and approaches to quantify this to create a business case for collaboration are mixed and varied, three main routes exist; training, operational utility (e.g. data, tools), or reputational enhancement. The latter works by differentiating the company from its competitors (i.e. more accurate risk
assessment through better science), providing arguments for retaining existing clients, and opening doors to new clients that sales teams can follow up. In reinsurance this can be more important than harvesting and protecting IP generated in collaborations.

Ways research scientists might provide support to their risk practitioner partner: These are mapped to the typology of impact
(i.e. practitioner benefit) in Sect. 5.2.2 using square brackets e.g. [2] and include a brief commentary on how and why benefits emerge. Suggestions are not ranked as utility will be case specific.

- *Undertake a literature review* e.g. comprehensive review of what is known about risks in an emerging peril-region such as Africa [4]. This is a safe (i.e. low risk), early stage deliverable if included into funding bids. It will appear
least like a burden to the academic if the subject is novel (i.e. publishable) and a likely impact (e.g. pending strategic decision) has been identified. It is time-efficient for the practitioner.
- *Deliver new research-based science* in the form of concepts or theories that can be implemented by the practitioner to operational advantage ahead of competitors, e.g. by engaging with the scientist in a co-designed project as the work progresses [1]. Feed-in could be by modifying a company's 'own view of risk', or by some adaption to their
catastrophe modelling process/model. When exploring ideas or methodological improvements at the cutting-edge (i.e. higher-risk), collaboration can be a low cost alternative for a practitioner as if sufficient novelty exists a substantial fraction of the cost might be supportable through public funding.
- *Develop a spreadsheet-based* 'decision support tool' [1]. Although this is too basic for most reinsurance users, it may be appropriate for some of their clients.
- *Provide training sessions* [3]. See list above.
- *Create software tool (e.g. in R-shiny)* associated with a statistical model developed during research [1,2,4]. Such accessible, interactive visualisations can raise awareness amongst internal management or external clients of saleable new functionality or product opportunities. Some practitioners encourage dissemination as supplementary material to a journal article.

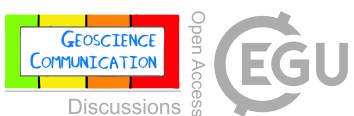

- *Develop a simplistic exposure-based risk model* (i.e. catastrophe model) of just sufficient complexity to illustrate a particular scientific insight (e.g. Royse et al., 2014). Rationale similar to the software tool.

- *Provision of expert advice* to some internal/external committee or decision-making group, perhaps *ad hoc* input on the latest science [2,4]. This is time-effective for the practitioner, and aligns with the academic's interests (see above).

- *Contributions to industry forums / conferences* (e.g. of RMS, AIR, Aon, Oasis) on co-designed work highlight a practitioner's engagement with the latest science [4].

- *Produce footprints for a catalogue of historical events* [1]. This is perhaps the easiest aspect of a catastrophe model for environmental scientists to contribute to. Other elements (e.g. vulnerability functions, stochastic event sets) either require sensitive data (i.e. claims) or to be fully benchmarked against industry standards before they could be operationalized.

- *Invite the practitioner to give a guest lecture (e.g. to undergraduates), seminar, or training.* A potentially enjoyable experience, an opportunity to discuss collaboration possibilities, and provides contact with students (e.g. PhD) who may apply for jobs with the company in future.

These lists are not, and do not attempt to be, exhaustive (e.g. *short placements* of ≲1 week). The key is open, honest and continual interaction based on an appreciation of motives, which may help to bridge frustrating gaps. Currently, provision of hazard footprints illustrates this; industry asserts a need for accessibility (e.g. on OasisHub) in an industry data format, and yet it is difficult for a researcher to prioritise doing this on only the speculation that impact may happen. With concrete and specific plans for creating and collecting detailed evidence of impact in place within the insurer, the academic may readily see the value in sacrificing research to do the work. Alternatively, a brief session to advise a consultant paid to undertake the work may be all that can be justified (e.g. see Moulin, 2018: p47) .

More widely, workshops (e.g. Dixon et al., 2017), brokerage events (e.g. for funding calls), industrially funded initiatives (e.g. Willis' Research Network, JBA Trust) and the use of distinct and separate middle people as translators/facilitators (e.g. consultants, NERC's KE Fellows) are noted, but are not focussed on here as they do not directly pertain to the motives of an individual academic whose 'core business' remains researching environmental science. Similarly, longer (i.e. ≲1 week) placements for either party (i.e. academic or practitioner) may also be very valuable but entail an extended absence from the employer and work environment (e.g. as described in Sect. 4) and are thus out of scope.

### 5.3.1 Summary of pragmatic ways to nurture collaboration

A trusting long-term relationship is vital, but has to start somewhere. In brief, a mixture of short-term steps (<1 year) to initiate a relationship, building toward longer-term and more substantive targets and outputs (1-5 years), is suggested. The



ideal is mutual-benefit at each stage; even if consultancy and giving advice are not what is ultimately critical to a university-based scientist (i.e. subject-leading publications), most are patient in developing towards this (e.g. industrial relationship and funding bids).

Since reinsurance businesses are typically multi-national, and academic motivations (e.g. funding, publications) are similar internationally, this advice is not UK specific as it assumes a pre-existing motivation towards impact. Furthermore, it may act as a template, a basis to design guides in other industrial sectors.

**6. A final comment: evidence of impact in practice**

This paper in itself provides an excellent illustration of the practicalities involved in creating benefit for an academic via evidence-based impact; even if you do change behaviour after reading it, no feedback into items of concrete value to the academic co-authors (i.e. appraisal, REF Impact case study) is possible without evidence. And, how will the university-based scientist find out if they are not told? Please remember that *Raising Awareness* is classed as impact (Reed, 2018) (Sect. 5.2.2), and even a 1-2 line e-mail to the authors (j.hillier@lboro.ac.uk) noting this and potential plans (e.g. *'I might now*

*consider using an academic for internal training'*) would class as evidence.

**7. Conclusions**

University-derived environmental science of value to industry is best obtained in genuinely co-designed projects and long-term collaborations, which demand the investment of significant time and effort to create and maintain. What then may motivate an academic to work with industry in this way? And, reciprocally, what might constrain them?

The main findings of this work are about the nature of the severe challenge posed by the heavily time-constrained culture of today's universities, mainly derived from UK data but at least in part applicable much more widely (e.g. Australia, Europe, Social Science, Engineering). Importantly, a tension in shown to typically exist between exciting curiosity-driven opportunities in university-business collaboration, and workload.  Thus, to justify the time to work with industry, the work

must inspire curiosity and facilitate future cutting-edge and world class science in order to mitigate the conflict with an academic's overriding imperative to publish. It must also provide evidence of real-world changes, and ideally other reportable outcomes (e.g. official status as an insurer's advisor), to feed back into the scientist's performance appraisals. A conceptual model (Fig. 3) is developed of the inter-relationships between factors in an academic job (e.g. teaching, impact, publications).

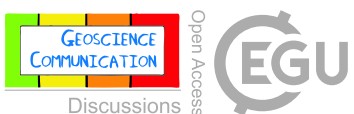

Although each university-based environmental scientist's motivations may differ dramatically due to a variety of factors (e.g. personality, work history, career stage, family commitments, current institution), a number of key common tendencies exist. It is well established that most scientists are driven by curiosity not additional personal financial reward, but this work also highlights the following points relevant to university-business collaborations:

- Amid a raft of 20-50 key duties, scientists may be able to free *up to* 0.5 days/week for work with risk practitioners (i.e. impact activities).
- In addition to career, desire to work with business and be useful (i.e. 'utility') and to aid society (i.e. 'altruism') are identified as intrinsic drivers for undertaking impact-related work.
- The conflicting roles required in academia do not yet well support the long-term investment of time needed for research impact when weighted against shorter-term pressures (e.g. annual appraisal, teaching delivery).
- Given the time limitations on both parties (e.g. academics and practitioners), it is necessary to establish coping strategies to secure initial traction and to build a relationship.
- A variety of pragmatic short-term (<1 year) steps are proposed for *a la carte* use to initiate and nurture a relationship. Explanation is provided of *how* these mitigate the dis-incentives within today's academic environment, align with industry needs, and contain the potential for mutual benefit at each stage.
- Short-term steps (<1 year) to initiate a relationship, building toward longer-term and more substantive targets and outputs (1-5 years), are suggested.

20   This work raises a number of related questions for future study. Pricing structures and tight deadlines can limit academics ability to engage with contract work. So, how might academics best engage with existing consultancy firms? Once an academic has formed an industrial relationship, a university might want to use it for its own wider purposes. So, who 'owns' a university-industry relationship, and how might they be best managed? It is hoped that the work in this paper provides part of the basis for an up-to-date approach to these questions.

**Data availability**

Metadata (e.g. web links) for the 20 sets of documents for UK job specifications and promotion criteria are given in Supplementary Material, but are not supplied for copyright reasons. Personal data for the KEN participants is not provided or retained, but derived data are given within the manuscript (i.e. Table 1) or as supplementary material i.e. for the word clouds (Figs. 1,2).

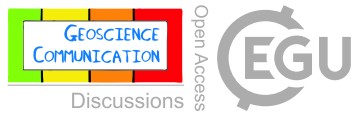

**Author contributions**

JH designed the study, led data collection and undertook initial drafting of the manuscript. All authors contributed observational, experience-based data about academics' behaviour and/or insurance-based practice, and to the writing of the manuscript.

5   **Competing interests**

The authors declare that they have no conflict of interest.

**Acknowledgements**

JH was funded by NERC grant NE/R003297/1. We thank the attendees at NERC's KEN workshop in June 2018 for participating. Mark Reed's comments much improved an early version of this manuscript, as did James Esson's on a later
10   version.


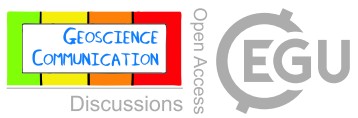

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



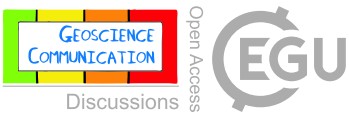

Table 1: Illustrative expectations of a typical early- to mid-career UK academic, based on a thematic analysis. Numbers in brackets (e.g. [7]) indicate the number of occurrences within 10 job specifications. Square brackets [] are for analysis by the lead author, and curved brackets () from the KEN workshop, which agree well (i.e. $R^2 = 0.77$). Grey text distinguishes items not in the majority of job specifications.

| Category | Tasks |
|---|---|
| 1. Research<br><br>(2 days/week) | • Networking (e.g. internal to international), seminars, and unfunded initial studies to define and initiate potential projects **[8] (9)**<br>• Preparing external funding bids, building multi-institutional teams including external stakeholders (e.g. insurers) **[9] (10)**<br>• Competing for internal funding (e.g. for PhD students or pilot studies) **[0] (1)**<br>• Management of any funded grants (e.g. finances, line management of researchers) **[4] (4)**<br>• PhD supervision **[7] (4)**<br>• Reviewing papers and funding bids written by others **[2] (2)**<br>• Presenting at and organizing conferences (e.g. designing & implementing sessions) **[5] (3)**<br>• Own hands-on research, including learning any new skills required and any associated reading of journal papers **[10] (8)**<br>• Writing own (or co-authored) peer-reviewed journal articles **[9] (9)**<br>• Own impact-related work **[6] (3)** |
| 2.Teaching<br><br>(2 days/week) | • Undergraduate large-group teaching in lectures, practicals, field classes etc .... including design and delivery of all material, maintenance of an electronic learning system and all student contact (e.g. discussions, e-mail queries, formative feedback). **[10] (8)**<br>• Undergraduate skills-based tutorials, dissertation supervision, pastoral care and follow-up contact (e.g. job references) **[7] (8)**<br>• Setting and marking of assessments (e.g. exams, fieldwork exercises) **[8] (8)**<br>• Postgraduate level teaching, mirroring the undergraduate requirements. **[7] (6)**<br>• Pedagogical research or self-reflection to innovate teaching delivery (e.g. creating simulation tools for interactive interludes during lectures) **[8] (7)**<br>• Continuing Professional Development courses relating to teaching **[3] (1)** |
| 3. Administration<br><br>(1 day/week) | • As convener of taught modules, logistics (e.g. rooms, equipment, personnel). **[5] (5)**<br>• Various contributions to departmental functions; illustratively, recruitment (e.g. open days), committees (e.g. teaching and learning, strategic planning), PhD student related (e.g. progress review and examination). **[8] (8)**<br>• Sundry (e.g. appraisals, expenses) **[2] (3)**<br>• Skills training (e.g. project management, recruitment skills) **[3] (2)**<br>• Typically, also a significant administrative role (e.g. Admissions Tutor, Programme Coordinator, Health & Safety Officer) **[9] (6)** |



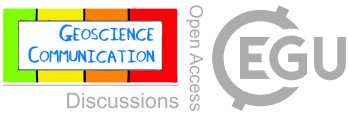

Table 2: Indicative set of appraisal criteria for an early to mid-career research scientist based in a UK university, as distilled into promotion criteria to Senior Lecturer (approx. equiv. to Associate Professor). Percentages indicate the relative occurrence of the categories as main headings within the criteria.

| Category | Indicative examples |
|---|---|
| Research [100% i.e. 10 of 10 sets of criteria] | 1. Role model of good practice in PhD supervision, with successful completions <br> 2. Established international reputation <br> 3. Evidence of a strong, independent research profile and programme (e.g. excellent and sustained record of publications) <br> 4. Successful in securing external grant funding |
| Teaching [100%] | 5. Fellow of the Higher Education Academy (i.e. attain teaching qualification) <br> 6. High-quality and well-received delivery of stimulating and distinctive undergraduate and postgraduate level teaching, <br> 7. Innovations in delivery, or leading in policy and practice, or strategic developments (e.g. to programmes) |
| Enterprise/Impact [30% i.e. 3 of 10] | 8. Consultancy, or other income-generating work (e.g. starting a spin-off company, exploring atypical funding opportunities) <br> 9. Engagement with the wider world (e.g. collaboration, media, policy) that has significant and demonstrable impact (e.g. suitable for a REF Impact Case Study). |
| Leadership/ Administration [100%] | 10. Leading internally and developing leadership outside the institution <br> 11. Sustained success and innovation within a significant managerial/administrative role |

