# Peer review of "Demystifying academics to enhance university-business collaborations in environmental science"

_Geoscience Communication, 2018_

## Referee Comment (RC1) · Dr Westaway (Referee) · 11 Sep 2018

**Review of "Demystifying academics to enhance university-business collaborations in environmental science" by Hillier et al.**

Summary

This is a thoughtful and well written paper that attempts to shed light on both what motivates and constrains a typical UK-based early- to mid-career environmental scientist, and uses this intelligence to help understand how industry-university collaborations might be enhanced. The co-authors draw on their first-hand experiences, supplemented by carefully selected textual data and the outcomes of a participatory workshop. Perhaps most usefully, the paper presents two lists of practical and short-terms measures through which industry practioners can develop a relationship with an academic partner and *vice versa*.

General comments

1. **Context**: The paper gives no real context in terms of how (UK) environmental scientists currently engage with industry partners (i.e. collaborate, receive funding), both in terms of proportion (how common it is) and typical time commitment (how much time is dedicated). Such an assessment, even if brief, would provide a useful illustration of the current situation. While I am not aware of any references that have directly collected such data, there may be other (perhaps more quantitative) strategies that provide a sufficient proxy measure (e.g. proportion of NERC grants awarded that have included industry partner, proportion of published papers in a specific discipline that include industry partner as co-author or industry funding), perhaps with direct approaches to the academics in question to gather a ballpark estimate of 'average' time commitment. Some 'top-down' assessment such as this would provide an interesting (and hopefully corollary) counterpoint to your current largely 'bottom-up' approach.

2. **International perspective**: The paper attempts to include an international perspective on many of the issues covered, but I felt at times that this disrupted the main narrative of the paper. I would suggest that the occasional paragraphs discussing international perspectives are removed; I do not believe this would impact on the quality of the paper as it is evidently and usefully a UK-focussed account. Alternatively, there could be a dedicated section towards the end of the paper that provides all international perspectives in one place (rather than them being dispersed through the paper).

3. **Part-time academics:** The environmental scientist persona is (implicitly) a full-time post. While the majority of UK academics are full-time, there is a small but significant proportion of academics that work part time (~33%, see https://www.hesa.ac.uk/news/18-01-2018/sfr248-higher-education-staff-statistics). Part-time researchers would be expected have a reduced range of responsibilities commensurate with their reduced hours, but this might be expected to limit their time available for industrial collaboration to an even greater extent than for full time staff. It

would be useful if you could provide a brief commentary on the additional issues surrounding part time academic contracts, and if they were considered as part of this study?

4. **(Conceptual) pie chart of time constraints:** On p17, lines 23-28 you discuss prioritisation of time and that adding a new task requires discarding of an existing one. I feel this could be a very useful concept more widely in your discussion of time constraints. An academic does not have an endless list of tasks (as your table perhaps implies), but rather has finite time (i.e. a pie chart) that must be prioritised into different activities (segments of the pie). It would perhaps be informative if you were able to present the information shown in Table 2 as a pie chart, at least conceptually, with different segments representing the competing demands on a academics' time. This should illustrate more clearly how only 0.5 days per week are available for industrial collaboration, or how finding time for new collaborations necessarily requires other tasks being reduced in time or omitted.

5. **Access to published work:** On p20, lines 1-5 you advise industrial practioners to check "that there isn't already an answer to your question in the research literature?". From my experience, this has been (and remains) a barrier between industry and academic research as industrial practioners in many fields do not routinely have access to academic journals behind paywalls. This both prevents them reading about new science and also reduces their incentive to get work published (as their peers would not necessarily see or benefit from it). The rise of open access publishing is changing this, but some discussion of the how the traditional academic publishing model can create a divide between universities and industry might be helpful in this context.

6. **Academic impact – improved methods/data/results:** On p21, lines 15-33 you outline some of the types of impact that can be demonstrated. A notable omission from this list is how academic involvement can improve the methods/data/models/results being used by industry practioners. Although this ultimately may lead to increased profitability, the other benefits of such improvements (e.g. time savings, reputational benefits) are such that I believe it deserves to be listed as a specific impact in your list. In your insurance setting, examples might include improvements to the flood model, baseline topography or property database being used to assess properties at risk.

7. **Academic impact – time lag:** There is no discussion of the fact that while a published paper can be 'recorded' immediately, there is often considerable time-lag before other impacts can be properly assessed (e.g. time to implement changes across a business, time to (re)train staff, time to quantify cost savings), which can be hard to reconcile with in some cases relatively short term academic decisions/appraisals/funding/etc.

8. **Motivation of industry:** There is little commentary of what industry are looking for from collaboration with universities. Although the stated purpose of the paper is to give insights from the academic's perspective, it is hard/impossible to divorce this completely from the motivations and needs of industry. From my experience, the nature of industry requirements in some cases can be (perceived as) such that there is limited appetite for research scientists to engage, even if they did have the time (e.g. low-risk non-innovative solutions, short time timescales, pressure to produce results).

9. **Institutional-scale responses:** The paper generally focuses on collaborations that are established and maintained at individual-scale. However, there are university/department-scale responses that are intended to facilitate industrial collaboration (e.g. dedicated 'industrial liaison officers' such as www.bristol.ac.uk/engineering/ilo/academics/, industrial-funded academic positions, funding institutional facilities, etc). It would be good to have some discussion of if and how such initiatives influence or change the papers' findings.

Specific comments

10. **p3, line 4:** I would suggest that it is "knowledge" that is produced rather than "science".

11. **p4, line 26-28:** It is not clear to which reference the gold/puzzle/ribbon classification is being attributed (Lam 2011 or Stephan and Levin 1992).

12. **p7, line 28:** Suggest that sect 2.1.2 does not start with a question. I suggest this question is either simply deleted or rephrased into a statement.

13. **P9, line 5:** "a guide for academics to their academic partner" – it is not clear to me what this statement means?

14. **p11, line 29:** "..only a sub-part of this." This is tautologous; I suggest "subset" or "part"

15. **p13, line 1: ".., then winning... "** – This would read better as ".., followed by winning..."

16. **p16, line 2:** "trusting" – This is perhaps better described as "trust-based"

17. **p16, line 14:** "65-89% of university scientists..." – this seems are very precise range, which stands out all the more as it starts the paragraph. It might be better to state a more descriptive range here, e.g. "At least two thirds..." or "As many as 90%..."

18. **p17, line 3:** "...days and days to sit, gazing around and pondering..." – this feels too colloquial. Perhaps "...significant 'thinking time'..." or "...time to engage in deep thought on blue sky research questions.

19. **p18, line 5:** "...from tight expectations..." – "strict" or "rigid" might be a better word than "tight" here

20. **p18, line 14-15:** "Thus, there is some need for continuity, which may be perceived as 'pet projects' by industry. – It is not clear what is meant by "pet projects" and no further explanation is given? And why is this a problem? A contrary perception is that it creates world leading scientists in their specific field.

21. **p19, line 8:** "...a scale they could otherwise only dream of?" – Rather colloquial. Suggest "...a scale not otherwise possible?"

22. **p23, line 5:** "buy a post-docs time" – Colloquial. Suggest same phraseology used as on p13, line 2 ("by funding a post-doctoral researcher").

23. **p24, figure caption**: "...and 3 main motivations after 'gold' discounted are in italics" – I do not understand what is meant here? I can only see two further motivations in italics ('utility' and 'altruism'). I would like to see how 'puzzle' and 'ribbon' map onto these relationships too.

24. **p24, line 9:** See comment 13

25. **p26, line 11**: While I agree with the sentiment, "Brilliant!" is not the best way of starting/explaining this step. Can this be reformulated as a statement/sentence (e.g. "While this is perhaps the ideal scenario for many academics, the sums....").

26. **p27, line 14**: "Ways research scientists might provide support to their risk practitioner partner:" – I am not clear whether this is meant to be a sentence (in which case it should be reworded) or a section heading (in which case it needs re-formatting)

Technical comments

27. **"ise" or "ize":** There is some inconsistency between these two spellings though the manuscript.

28. **p4, line 26**: Punctuation needs correcting "...into , 'gold'...".

29. **p13, line 4:** Missing "are" between "outcomes" and "also"

30. **p14, line 5:** Missing apostrophe after "academics"

31. **p14, line 9:** Should be "nor" rather than "or"

32. **p16, line 13:** Delete apostrophe.

33. **p21, line 6:** "UK's 7 yearly" should be "UK's 7-yearly"

---

## Referee Comment (RC2) · A. Mackay (Referee) · 12 Sep 2018

The study provides a model and pathway to increase academic participation in industry, by outlining (i) a detailed analysis of the time constraints of an early career academic, and (ii) a set of incentives that might encourage this academic to work with industry.

The first part of the study provides an overview (for I assume a potential collaborator in industry) of working conditions and time constraints of a typical academic. I have some reservations, however; overall it could be more succinct, not include links to out of date or wrong information, and not perpetuate poor practise in academia. The second part lays out how people in industry can try and work with academics, with a view to looking at ways to make collaborations of interest to academics and fruitful. This part of the

manuscript is well written and more successful in terms of communicating the potential synergies between industry and academia. However, the emphasis of risk between any collaboration is weighed towards the academic rather than the industry partner. For example, P26, Fund blue skies research. The conclusion here seems to be that to fund a PhD is risky for the industrial partner as there is a "chance of failure". Blue-skies research by its very definition is risky, and joint PhD students are one of the best ways to link industry and universities together, as is recognised by many UKRI initiatives e.g. the Industrial Doctoral Centres funded by the ESPRC, NERC Industrial Case Students, NERC Industrial Strategy Innovation Placements etc.

But my main concern with the manuscript as a whole is with the figure of 0.5 days per week, calculated to be the amount of time an "efficient and effective" academic could set aside to work with industry. The figure really just appears, and is not based on a critical analysis of the empirical data as far as I can tell (although seems to be derived from Table 1). Also, it is based on the authors accepting that working a 50h+ week is acceptable, which I challenge below.

Overall, the article could be more focussed on just the UK system, and just on the environmental and geosciences. A summary of the extent of existing academic-industry collaboration would be helpful, and how the geosciences / environmental fields contribute to this. This would give some needed background asto the extent of the issue. Throughout the manuscript, examples to e.g. international centres or practises are given, but these are far too few and not comprehensive, so their added value is low, and only detracts from the key take home message about UK collaborations.

The article could discuss more implications of the current external drivers for research and teaching, such as REF and TEF. NSS is mentioned on P13, and TEF once on P14, but the URL given is now out of date as HEFCE no longer exists. REF impact is mentioned, but that this is such a significant driver of potential collaboration between academics and industry, it is a missed opportunity not to discuss this more. Also, I would personally like to have seen greater consideration of equality and diversity

issues related to potential collaboration with industry; not only do these considerations tie into employment structures such days or hours worked per week, but also have implications for potential funding sources, e.g. if your department has Athena Swan recognition etc.

1. Does the paper address relevant scientific questions within the scope of GC?

Yes – successful communication between academics, industry and business is an essential component of universities contributing to the economic wealth of the country. However, I would have liked to have seen a stronger case being made for the need for this knowledge. UK universities are arguably very successful in collaborating with industry and business, and while tensions for time will always exist, is this really new? Comments such as those on P3, Lines 17-18, starting "By better understanding..." are fine, but this study is aimed at early career scientists, who may not yet be "world-leading"; this is an important distinction in terms of expectations of knowledge, resources etc.

2. Does the paper present novel concepts, ideas, tools, or data?

The study from the outset suggest that the novel concept, or unknown parameter, is how academic workloads and incentive structures may act as a barrier to industrial-academic collaborations. Given the success of university – industrial collaborations in the UK already, I wonder if the first really is a barrier. It could be in the geosciences / environmental sciences, and maybe that needs to be explored a bit more deeply than university-industry collaborations in general. The wealth of existing training schemes, placement schemes, success of spinout companies, contributions of UK academia to GDP etc all suggest that issues with time constraints are well known but already workable with.

The statement, which in a sense is a crux of this study, "...there has been limited attention devoted to the exact nature of barriers facing academics..." needs to be evidenced. Some quick on-line searches shows that there are numerous studies looking
at the barriers academics face in terms of forging university-industry links. In fact, research intensive universities will have whole teams dedicated to addressing these challenges.

3. Are the scientific methods and assumptions valid and clearly outlined?

I do have some issues with aspects the methodology undertaken. The study paints a picture of a 'typical' early career academic at the Senior Lecturer (SL) scale of their career. This is based on 10 job adverts (which I think is rather low). However, some aspects of the methodology are either wrong, or not reflective of the rapidly changing academic environment today.

For example, on P7, lines 18-20, the authors link to some generic guidance to job descriptions. However, The information given here is wrong and very out of date. The link provided is for academic-related (AR) jobs: these are staff categories that were once seen to support academic workings, but are now viewed as professional services (PS), i.e. careers in their own rights. Whether termed AR or PS, the guidelines linked to here are not for the academic (teaching and research) jobs being discussed here.

The following two points are more picky I suppose, but potentially important in understanding the expectations of an early career academic at the SL level. First of all, I don't agree with the conflation of appraisals and promotion criteria. The study suggests that appraisals are "a relatively new phenomenon", but they are not; they have been undertaken at universities for over 20 years. Moreover, they are certainly not undertaken to "judge" (P7, line 31). Appraisal are designed for academics to set out aims and objectives for undertaking their job on an annual basis, and to have discussions as to whether these have been met. Promotion criteria on the other hand request for feedback on one's reputation for mainly research and (teaching) scholarship, so to bring the two together is not particularly helpful as they work on very different timescales. Second, the study is based around academics at the SL level, but suggests on P9, line 29 that this is equivalent to the North American Associate Professor (AP), but I would
argue that AP was more equivalent to Reader, where a UK academic is recognized for their world-leading research, as would someone be in the States on being awarded tenure and AP. For a SL, teaching and scholarship plays a stronger role in evaluations.

For me, more problematic, is "de facto expected" number of hours an SL is expected to work at week, up to 50 hours or more. It describes working 10h days or working at weekends as "respectable length". These practices are greatly at odds with moves to have greater equality and work-life balance that reflects the needs for academics having caring responsibilities, to have a life outside of academia, and to minimize stress and mental health issues. The study goes on to state that the ideal person "is in good mental health" and are "efficient and effective" in their approach to research, to the point that "they would not remain in the their position if they were not" (P10, lines 1-2). This is patently nonsense – academics are not removed from their jobs on the basis if they are efficient or not. For me this gives the impression that the model being developed - for a person to devote 0.5 days a week to collaborate with industry - will not work if an academic does not want to, or cannot work, a 50h week, or miss time with families at weekend, or experience any kind of mental health issues etc. These factors are not trivial – they have important implications for diversity and equality issues, and personally, I would question the value of such a model from the outset.

Finally, the assumption that [P13, line 20] "PhD students can be an effective means to generate publications in a time-limited university environment" is not one that I or any academic I know would condone, and has no place in modern day academia.

4. Are the results sufficient to support the interpretations and conclusions?

Not really. The conclusion seems to be that an efficient and effective academic can probably find 0.5days a week to collaborate with industry, but I question the data and assumptions that this is based upon. It seems to me that this figure is a qualitative amount that the academics in the cohort suggested that they may be able to find, out of their already busy schedules. Which is fine, but this does not seem to be derived

from a critical analyses of what a SL does on a day to day basis.

5. Do the authors give proper credit to related work and clearly indicate their own new/original contribution?

I think so.

6. Does the title clearly reflect the contents of the paper?

The title mentions "business", but in the manuscript refers mainly to "industry"

7. Does the abstract provide a concise and complete summary?

This was fine.

8. Is the overall presentation well structured and clear?

I thought that the Introduction, setting the scene etc could be much more succinct. Many times the manuscript refers the reader to future sections as justification of what is being stated, but this made reading of the paper difficult, as you have to keep going backwards and forwards to find out what is being referred to.

9. Is the language fluent and precise?

Overall it is fine. But there were a few tropes that could be avoided, such as P17, lines 2-3 "So, this work confirms that an academic with days and days to sit gazing around and pondering is a myth...". This suggests a misunderstanding of how an academic may approach their writing, rather than just doing it. Writing does require thinking and pondering, for days, sometimes weeks or months, so I'm not really sure what is being got at here.

10. Are the number and quality of references appropriate?

These look fine.

---

## Author Comment (AC1) · 27 Nov 2018

**Demystifying academics to enhance university-business collaborations in environmental science**

We thank the two *Geoscience Communications* reviewers (R1 & R2) for their thoughtful comments. We have also received an internal review from a Social Science (Human Geography) colleague at Loughborough (R3). This is formal in the sense that it was written and submitted for university evaluation purposes.  Thus, in the spirit of open review, we also provide these comments and responses to them.  However, to avoid any conflict of interest, this is given lower precedence than *Geoscience Communication's* external reviews.

Main responses:

- R1 & R2 would like to see more clearly how the figure of *up to* 0.5 days per week for industrial collaboration derives from the data. We have added a pie chart as suggested by R1, and a longer critical discussion of how the figure arises from the various data sources along with the uncertainty in this estimate as suggested by R2 is now in Sect 6.1.1.
- R1 & R2 suggest a partial restructuring of the manuscript. As suggested by both, we have removed the material on international interest from where it is interlaced with the main text, and created a new section at the end as proposed by R1.  The Introduction is simplified and focussed on defining the academic research question, with the insurance sector case study now in its own section.
- R3 would like a less tentative tone about the novelty and significance of the methodological design and new conceptual model.  This has been done.
- Extra context and information has been added in a number of places (e.g. on UK environmental scientists' current engagement with industry for R1), but this is kept as succinct as possible to prevent the manuscript lengthening too much.

We have responded individually and in detail to all the other comments (e.g. minimizing forward referencing, decoupling ~50h/wk from <0.5 days and the conceptual model). Reviewers' comments are in grey, with responses in black. Line and page numbers in responses (e.g. P4L7) correspond to the revised text, and a Word document with track changes is also provided for ease of evaluation. Individual comments are identified by reviewer and comment number where necessary (e.g. R2C5).

**Reviewer 1 – Dr Richard Westaway**

*Summary*

This is a thoughtful and well written paper that attempts to shed light on both what motivates and constrains a typical UK-based early- to mid-career environmental scientist, and uses this intelligence to help understand how industry-university collaborations might be enhanced. The co-authors draw on their first-hand experiences, supplemented by carefully selected textual data and the outcomes of a participatory workshop. Perhaps most usefully, the paper presents two lists of practical and short-terms measures through which industry practitioners can develop a relationship with an academic partner and vice versa.
> Thank you.

*General comments*

1. Context: The paper gives no real context in terms of how (UK) environmental scientists currently engage with industry partners (i.e. collaborate, receive funding), both in terms of proportion (how common it is) and typical time commitment (how much time is dedicated). Such an assessment, even if brief, would provide a useful illustration of the current situation. While I am not aware of any references that have directly collected such data, there may be other (perhaps more quantitative) strategies that provide a sufficient proxy measure (e.g. proportion of NERC grants awarded that have included industry partner, proportion of published papers in a specific discipline that include industry partner as co-author or industry funding), perhaps with direct approaches to the academics in question to gather a ballpark estimate of 'average' time commitment. Some 'top-down' assessment such as this would provide an interesting (and hopefully corollary) counterpoint to your current largely 'bottom-up' approach.
> As R1 recognises, data to evidence engagement between researchers and industry are not collated in any readily accessible form; we contacted both NERC's 'Innovation' and 'Evidence' teams to establish this more

firmly. The best summary might be Dowling's 2015 study (points 17-25), which had sufficient resources to commission an analysis of all REF case studies, and to question all UK universities for data, but even then were able to draw what are really very limited conclusions from the 12,240 case collaborations reported to them i.e. collaboration is 'very patchy' and, tentatively, that there are a some companies that are very active in building research collaborations with universities whilst a large number collaborate in a relatively restricted way with universities.  P6L10

> In terms of time commitment (i.e. time dedicated to impact), the results of work reported in this manuscript offer some initial information.

> By creating a separate section on the insurance sector, and re-ordering the material previously in the Introduction, we present what little context is currently known from the government/NERC side (i.e. Dowling [2015] and  Goff [2015]) whilst placing it in the context of an overview of how environmental science is used and produced at the moment by the insurance sector. P5L31-P6L7.

> In summary, we agree with the reviewer and Dowling (2015) that a better understanding of the collaborative landscape in the UK is desirable, and look forward to seeing research into it, although that will be a substantial undertaking.

2. International perspective: The paper attempts to include an international perspective on many of the issues covered, but I felt at times that this disrupted the main narrative of the paper. I would suggest that the occasional paragraphs discussing international perspectives are removed; I do not believe this would impact on the quality of the paper as it is evidently and usefully a UK-focussed account. Alternatively, there could be a dedicated section towards the end of the paper that provides all international perspectives in one place (rather than them being dispersed through the paper).

> R1 agrees with part of comment from Reviewer 2 [R2C1]. Throughout the paper, the international perspective has been moved to a separate section at the end of the discussion (new Sect. 7) P31L12.  This section now highlights the global applicability of the results and discusses where there may be differences to the UK system. An emphasis on Australia arises because it, like the UK, is at the forefront of integrating impact into the university sector, and because the limited academic work in this area is focussed on Australia.

3. Part-time academics: The environmental scientist persona is (implicitly) a full-time post. While the majority of UK academics are full-time, there is a small but significant proportion of academics that work part time (~33%, see https://www.hesa.ac.uk/news/18-01-2018/sfr248-higher-education-staff-statistics). Part-time researchers would be expected have a reduced range of responsibilities commensurate with their reduced hours, but this might be expected to limit their time available for industrial collaboration to an even greater extent than for full time staff. It would be useful if you could provide a brief commentary on the additional issues surrounding part time academic contracts, and if they were considered as part of this study?

> Thank you for this comment. 'full-time' added to the abstract to make the 0.5 days/week figure explicit, and a brief commentary about scaling this to part-time staff has been added in Sect. 6.1.1, although the issues around part-time contracts were not within the scope of this study. Note that the lead author is part-time (0.8 FTE). P19L1

4. (Conceptual) pie chart of time constraints: On p17, lines 23-28 you discuss prioritisation of time and that adding a new task requires discarding of an existing one. I feel this could be a very useful concept more widely in your discussion of time constraints. An academic does not have an endless list of tasks (as your table perhaps implies), but rather has finite time (i.e. a pie chart) that must be prioritised into different activities (segments of the pie). It would perhaps be informative if you were able to present the information shown in Table 1 as a pie chart, at least conceptually, with different segments representing the competing demands on an academic's time. This should illustrate more clearly how only 0.5 days per week are available for industrial collaboration, or how finding time for new collaborations necessarily requires other tasks being reduced in time or omitted.

> Thank you for this suggestion. We have added a pie chart relating to Table 1 (new Fig. 3) in the discussion in order to more clearly illustrate how only up to 0.5 days per week are available for industrial collaboration.  P18

5. Access to published work: On p20, lines 1-5 you advise industrial practitioners to check "that there isn't already an answer to your question in the research literature?" From my experience, this has been (and remains) a barrier between industry and academic research as industrial practitioners in many fields do not routinely have access to academic journals behind paywalls. This both prevents them reading about new science and also reduces their incentive to get work published (as their peers would not necessarily see or benefit from it). The rise of open access publishing is changing this, but some discussion of the how the traditional academic publishing model can create a divide between universities and industry might be helpful in this context.

> We have added slightly to the text so that we now explicitly acknowledge that access to publications may be a barrier (P21L30), but since we link to a way around this we prefer to avoid extending the text significantly with a discussion on the traditional academic publishing model.

6.  Academic impact – improved methods/data/results: On p21, lines 15-33 you outline some of the types of impact that can be demonstrated. A notable omission from this list is how academic involvement can improve the methods/data/models/results being used by industry practitioners. Although this ultimately may lead to increased profitability, the other benefits of such improvements (e.g. time savings, reputational benefits) are such that I believe it deserves to be listed as a specific impact in your list. In your insurance setting, examples might include improvements to the flood model, baseline topography or property database being used to assess properties at risk.

> The provision of improved *'operational utility'* (e.g. via data and tools) is now included in the first bullet point here, with this term taken from industry terminology used in Sect. 6.3 (P23L9). We have also added *operational utility* as a separate bullet point at the end of the list (P24L1), giving the types of evidence that may be effective for these impacts that are also pathways to impact (Reed, 2018). We have also included a reference in Sect. 6.3 back to this list (P29L17), and included *reputational enhancement* for full consistency between the sections (P23L31).

7.  Academic impact – time lag: There is no discussion of the fact that while a published paper can be 'recorded' immediately, there is often considerable time-lag before other impacts can be properly assessed (e.g. time to implement changes across a business, time to (re)train staff, time to quantify cost savings), which can be hard to reconcile with in some cases relatively short term academic decisions/appraisals/funding/etc.

> Whilst there is no extended discussion of this, it is designed into our practical hints and tips (i.e. Sect 6.3), noted explicitly at the end of the first paragraph (P26L20) there and bullet on *co-design of a research proposal*. Many assessments or appraisals of impact are initially focussed on 'potential impact' in recognition of the common time lag of impacts from research. In order to limit the size of the manuscript and because it does not alter the overall findings of the paper, we respectfully propose that we do not add anything more about this.

8.  Motivation of industry: There is little commentary of what industry are looking for from collaboration with universities. Although the stated purpose of the paper is to give insights from the academic's perspective, it is hard/impossible to divorce this completely from the motivations and needs of industry. From my experience, the nature of industry requirements in some cases can be (perceived as) such that there is limited appetite for research scientists to engage, even if they did have the time (e.g. low-risk non-innovative solutions, short time timescales, pressure to produce results).

> We agree that it is hard to divorce industry motivations from the paper's stated purpose of giving insights from an academic's perspective. The paragraph and second bullet pointed list in Sect 6.3 provides a list of suggestions on the ways that research scientists can support risk practitioners, which we feel provides some insight into the motivations of industry (P29L8). In addition, before publication, we intend to add a link to an upcoming briefing note 'Funding Science for Natural Hazards Insurance' that we are currently finalizing, which reports data to outline the process of decision-making in (re)insurance sector using environmental science, and so starts to provide understanding on what industry want. This addition will provide a link to the reader who wants to find out more about industry motivations it P5L25.

9.  Institutional-scale responses: The paper generally focuses on collaborations that are established and maintained at individual-scale. However, there are university/department-scale responses that are intended to facilitate industrial collaboration (e.g. dedicated 'industrial liaison officers' such as www.bristol.ac.uk/engineering/ilo/academics/, industrial-funded academic positions, funding institutional facilities, etc). It would be good to have some discussion of if and how such initiatives influence or change the papers' findings.

> Thank you for this suggestion.  We have used it to improve the last paragraph before Sect 6.3.1, where other responses were mentioned before. P30L30

*Specific comments*

10. p3, line 4: I would suggest that it is "knowledge" that is produced rather than "science". Accepted, changed. P6L9

11. p4, line 26-28: It is not clear to which reference the gold/puzzle/ribbon classification is being attributed (Lam 2011 or Stephan and Levin 1992). Text altered to clarify. P3L16

12. p7, line 28: Suggest that sect 2.1.2 does not start with a question. I suggest this question is either simply deleted or rephrased into a statement. **Rephrased. P9L25**

13. P9, line 5: "a guide for academics to their academic partner" – it is not clear to me what this statement means? **Apologies. This is a typo. Changed to "a guide for academics to their business-sector partner" P10L30**

14. p11, line 29: "..only a sub-part of this." This is tautologous; I suggest "subset" or "part" **The OED defines 'subpart' as *'a subordinate part of something'*. In this instance we are referring to a part of a part, so believe our usage is correct. However, this reviewer's suggestion [R1C4] to use pie chart (new Fig. 3) clarifies this better than dwelling on linguistic subtleties. Hypen removed in line with OED usage. P10L11**

15. p13, line 1: ".., then winning... " – This would read better as ".., followed by winning..." **Changed. P13L8**

16. p16, line 2: "trusting" – This is perhaps better described as "trust-based" **Changed. P16L6**

17. p16, line 14: "65-89% of university scientists..." – this seems are very precise range, which stands out all the more as it starts the paragraph. It might be better to state a more descriptive range here, e.g. "At least two thirds..." or "As many as 90%..." **Changed. P17L1**

18. p17, line 3: "...days and days to sit, gazing around and pondering..." – this feels too colloquial. Perhaps "...significant 'thinking time'..." or "...time to engage in deep thought on blue sky research questions. **Accepted. This phrase originates in an earlier version of this work intended to communicate to stakeholders, adding 'colour' and directly counteracting potential stereotypes about academics having 'spare' or 'free' time. This is now stated directly. P17L20**

19. p18, line 5: "...from tight expectations..." – "strict" or "rigid" might be a better word than "tight" here. **Text amended. P19L29**

20. p18, line 14-15: "Thus, there is some need for continuity, which may be perceived as 'pet projects' by industry. – It is not clear what is meant by "pet projects" and no further explanation is given? And why is this a problem? A contrary perception is that it creates world leading scientists in their specific field. **Term clarified and explanation added. P20L8**

21. p19, line 8: "...a scale they could otherwise only dream of?" – Rather colloquial. Suggest "...a scale not > x otherwise possible?" **Thank you. Changed. P20L32**

22. p23, line 5: "buy a post-docs time" – Colloquial. Suggest same phraseology used as on p13, line 2 ("by funding a post-doctoral researcher"). **Thank you. Changed. P13L9**

23. p24, figure caption: "...and 3 main motivations after 'gold' discounted are in italics" – I do not understand what is meant here? I can only see two further motivations in italics ('utility' and 'altruism'). I would like to see how 'puzzle' and 'ribbon' map onto these relationships too. **The italicization of the words 'Curiosity', 'Career' and 'Impact' has been increased to make them clearer. The mapping between the terminology in the figure and Lam's (i.e. 'puzzle' and 'ribbon') is given in Sect. 6.2, and has now been added to the figure. Also, the caption has been expanded to better explain the figure and allow it to be understood better standing alone from the main text. P24**

24. p24, line 9: See comment 13. **Sentence split in to two to improve readability. P26L13**

25. p26, line 11: While I agree with the sentiment, "Brilliant!" is not the best way of starting/explaining this step. Can this be reformulated as a statement/sentence (e.g. "While this is perhaps the ideal scenario for many academics, the sums....")? **Rephrased in line with R1's suggestion. P28L14**

26. p27, line 14: "Ways research scientists might provide support to their risk practitioner partner:" – I am not clear whether this is meant to be a sentence (in which case it should be reworded) or a section heading (in which case it needs re-formatting) **Changed to full sentence. P29L14**

*Technical comments*

27. "ise" or "ize": There is some inconsistency between these two spellings though the manuscript. **"ise" now used throughout.**

28. p4, line 26: Punctuation needs correcting "...into , 'gold'...". **Fixed. P3L17**

29. p13, line 4: Missing "are" between "outcomes" and "also". **Added P13L11**

30. p14, line 5: Missing apostrophe after "academics". **Added, but here intention is to refer to each individual academic, so placed before s.**

31. p14, line 9: Should be "nor" rather than "or". **Changed**

32. p16, line 13: Delete apostrophe. **No apostrophe on P16L13, so assumed referring to P17. Apostrophe is to show possession of a view of multiple people (i.e. co-authors), and is retained.**

33. p21, line 6: "UK's 7 yearly" should be "UK's 7-yearly". **Changed P22L32**

**Reviewer 2 – Prof. Anson Mackay**

The study provides a model and pathway to increase academic participation in industry, by outlining (i) a detailed analysis of the time constraints of an early career academic, and (ii) a set of incentives that might encourage this academic to work with industry.

1.  The first part of the study provides an overview (for I assume a potential collaborator in industry) of working conditions and time constraints of a typical academic. I have some reservations, however; overall it could be more succinct, not include links to out of date or wrong information, and not perpetuate poor practise in academia.

> More succinct: The first part of the study provides the evidence to create an overview of working conditions and time constraints for a typical academic. This (i) allows an improved model of academic behaviour to be described, with the purpose of (ii) creating a typology of driver to explain *why* particular actions within collaborations may be useful. If such a study already existed, this one could be more concise; this research gap is now more clearly described in the Introduction P3L9-34.  The wording has been considered throughout to make the text as succinct as possible.

> Out of date or wrong information: The 'wrong' information was purely auxiliary to one argument, and has been removed (see below), and the out of date link (i.e. HEFCE) was the only official one available at the time of submission even if we were aware it would soon be out of date. This has been replaced. P14L15

> Poor practice: We did not intend to appear to advocate poor practice, and have clarified the manuscript in this respect (see specific responses below)

The second part lays out how people in industry can try and work with academics, with a view to looking at ways to make collaborations of interest to academics and fruitful. This part of the manuscript is well written and more successful in terms of communicating the potential synergies between industry and academia.

> Thank you.

However, the emphasis of risk between any collaboration is weighed towards the academic rather than the industry partner. For example, P26, Fund blue skies research. The conclusion here seems to be that to fund a PhD is risky for the industrial partner as there is a "chance of failure". Blue-skies research by its very definition is risky, and joint PhD students are one of the best ways to link industry and universities together, as is recognised by many UKRI initiatives e.g. the Industrial Doctoral Centres funded by the ESPRC, NERC Industrial Case Students, NERC Industrial Strategy Innovation Placements etc.

> We agree that this bullet point is not the place for the comment about PhD students. It has been moved to the bullet on project co-design, where we have also included some UKRI initiatives. P28L15, P28L30.

But my main concern with the manuscript as a whole is with the figure of 0.5 days per week, calculated to be the amount of time an "efficient and effective" academic could set aside to work with industry. The figure really just appears, and is not based on a critical analysis of the empirical data as far as I can tell (although seems to be derived from Table 1). Also, it is based on the authors accepting that working a 50h+ week is acceptable, which I challenge below.

> We have adapted the manuscript to clarify that the figure of 0.5 days is not at all based on a 50+ hour working week. 0.5 days is a fraction, with the hours worked only used to establish time pressure (e.g. P17L20-24).

> The figure of 0.5 days per week does originate from the empirical data in Table 1 (21 participants synthesizing objective data), combined with data supporting and literature evidence of a time-limited working environment, and the evidence drawn from the cohort of 17 academic co-authors. The 0.5 days figure has been removed from the results, and an expanded analysis of how this arises is now in the Discussion (Sect. 6.1.1).  We now clearly state that the 0.5 days is an evidence-based yet indicative maximum figure. P17L31 At the end of Sect. 6.1.1, we now clarify that this is an estimate. P19L6.

> New Fig. 3 has been added to assist in clarifying the <0.5 days figure. P18

2.  Overall, the article could be more focussed on just the UK system, and just on the environmental and geosciences. A summary of the extent of existing academic-industry collaboration would be helpful, and how the geosciences / environmental fields contribute to this. This would give some needed background as to the extent of the issue. Throughout the manuscript, examples to e.g. international centres or practises are given, but these are far too few and not comprehensive, so their added value is low, and only detracts from the key

take home message about UK collaborations.

> The article is now focussed on the UK system. In line with a suggestion of Reviewer 1 [R1C2], international implications and applicability are now separated and considered at the end of the Discussion (new Sect. 7). Information on the extent of academic-industry collaboration which was in the Introduction is in now given in the new Sect. 2 '*Case study: Insurance sector*' including a recent NERC view. P4L1. We do not explicitly limit the first half of the study to environmental science any more than it is already because Reviewer 3 [R3C3b] recommends retaining generality and the existing literature on academic behaviour is not clearly partitioned this way (i.e. by subject).

3.  The article could discuss more implications of the current external drivers for research and teaching, such as REF and TEF. NSS is mentioned on P13, and TEF once on P14, but the URL given is now out of date as HEFCE no longer exists. REF impact is mentioned, but that this is such a significant driver of potential collaboration between academics and industry, it is a missed opportunity not to discuss this more.

> TEF is now mentioned 4 times, REF 12 including explicit mentions of Impact Case studies, and KEF twice, with some of these in a short discussion of the conceptual model in the context of these current internal drivers in Sect. 6.3 P25L9. In the discussion (Sect. 6.2.2) REF is still introduced in a section discussing impact as a motivator of collaboration, but we have strengthened the statement slightly to "[REF is] *….. a key mechanism to encourage effective collaboration*". P22L25

> Furthermore, implications of external drivers for research and teaching (albeit NSS & TEF/REF not explicitly mentioned) are central to the methodology (see bullet 1 in Sect 4.1). A key simplification used in this work is that appraisal criteria and job specifications are taken as a distillation of these external factors, explicitly interpreted by universities into a form that directly relates to the behaviour of academics (e.g. P8L4-5).

> The out of date link (i.e. HEFCE) was the only official one available at the time of submission even if we were aware it would be soon out of date. This has been replaced P22L10. A more up to date link about impact has been added to Sect 6.2.2 as well.

Also, I would personally like to have seen greater consideration of equality and diversity issues related to potential collaboration with industry; not only do these considerations tie into employment structures such days or hours worked per week, but also have implications for potential funding sources, e.g. if your department has Athena Swan recognition etc.

> As a part-time worker (0.8 FTE), who took shared parental leave recently and participates equally in childcare, the lead author (i.e. Hillier) has some interest in diversity and equality issues. For instance, this afternoon work on this response was not possible as 1 small child needed to be collected from nursery and looked after. However, this is at a tangent to the main focus of the manuscript, so we have only added limited further consideration of it (P19L1-5) although it would make an interesting and important follow-up study.

> We have altered the manuscript so that it better articulates our view of how collaborations might in the presence of life outside work (e.g. Sect. 3, P19L6-14), and now hopefully do not give the impression that we condone poor practice.

> We note that numerous issues exist with respect to encouraging collaboration in the context of diversity and equality. For instance, personally, it is frustrating that NERC's innovation placement schemes are effectively inaccessible to academics with young families as placements away from home are practically not possible. However, we feel that to treat this adequately would require proper focus, perhaps with additional data collection, and so is best placed in a future manuscript.

4. Does the paper address relevant scientific questions within the scope of GC?

Yes – successful communication between academics, industry and business is an essential component of universities contributing to the economic wealth of the country. However, I would have liked to have seen a stronger case being made for the need for this knowledge. UK universities are arguably very successful in collaborating with industry and business, and while tensions for time will always exist, is this really new? Comments such as those on P3, Lines 17-18, starting "By better understanding..." are fine, but this study is aimed at early career scientists, who may not yet be "world- leading"; this is an important distinction in terms of expectations of knowledge, re- sources etc.

> Please see response to comment 5 below to justify the novelty of, and need for, the work. The reasons for targeting early-career scientists is given in the 'Environmental Scientist Persona' section, but we believe that the eventual products of "by better understanding" (i.e. the practical tips in Sect 6.3) are likely to be more widely applicable.

5. Does the paper present novel concepts, ideas, tools, or data?
The study from the outset suggest that the novel concept, or unknown parameter, is how academic workloads and incentive structures may act as a barrier to industrial- academic collaborations. Given the success of university – industrial collaborations in the UK already, I wonder if the first really is a barrier.
> Although there is clearly a level of success of university-industry collaborations, pressure on academic time is established as a barrier (e.g. ranked 3 of 10 in Fig. 12 of Dowling, 2015).
It could be in the geosciences / environmental sciences, and maybe that needs to be explored a bit more deeply than university-industry collaborations in general. The wealth of existing training schemes, placement schemes, success of spinout companies, contributions of UK academia to GDP etc all suggest that issues with time constraints are well known but already workable with.
> As we note (see also response below) time constraints are well-known as a barrier, and there are evidentially/anecdotally barriers to collaboration in the geosciences (e.g. with insurance where NERC issued a directed call for a KE Fellow to help bridge the gap between science and industry). What is less well known (and not academically reported on) is exactly *how* the barrier operates or ways to overcome it. This paper tries to understand the barrier's operation, and by doing so is able to propose practical suggestions to work around it.
The statement, which in a sense is a crux of this study, "...there has been limited attention devoted to the exact nature of barriers facing academics. . ." needs to be evi- denced. Some quick on-line searches shows that there are numerous studies looking at the barriers academics face in terms of forging university-industry links. In fact, research intensive universities will have whole teams dedicated to addressing these challenges.
> The offending sentence has been modified. P3L6
> This sentence was an inadvertent overstatement, created during re-writes and perhaps associated with an over-complex structure for the introduction. 'Exact nature' was originally written to link to investigating the exact nature of how motivations interact with incentive structures in the context of understanding how impact integrates into an academics work life, leading to practical tips to surmount barriers; this link was then broken.
> The research gaps (i.e. (i) how impact fits with models of motivation, and (ii) exactly how time acts as a constraint in a context of motivations/appraisals) are now more clearly positioned in the paragraphs immediately following (P3). These are followed through the paper e.g. into the conclusions P33L6

6. Are the scientific methods and assumptions valid and clearly outlined?
I do have some issues with aspects the methodology undertaken. The study paints a picture of a 'typical' early career academic at the Senior Lecturer (SL) scale of their career. This is based on 10 job adverts (which I think is rather low).
> The study uses 10 job adverts and 10 sets of appraisal criteria, which are then melded with and contextualised by the knowledge and experience of 17 academic co-authors and 21 participants at KEN workshop (albeit with some overlap between these sets as noted in the paper), in total providing information from 36 universities even if only participants/co-authors current positions considered.  Although obviously less that internet-based questionnaire surveys (e.g. Abreu, 2009; Bothwell, 2018), this compares well with the well-regarded social science studies in the literature about academic motivations; for instance, 35 interviews from 5 universities + online questionnaire (Lam, 2011), 1 university (Cadez, 2017), 25 interviews at 1 university (Harland, 2018), 30 collaborations (Frietas, 2017).
> Regarding Table 1 in particular, we argue that the emergence of a robust signal in itself indicates that 10 adverts is sufficient in the context of scrutiny from participants and co-authors. P9L23

However, some aspects of the methodology are either wrong, or not reflective of the rapidly changing academic environment today. For example, on P7, lines 18-20, the authors link to some generic guidance to job descriptions. However, The information given here is wrong and very out of date. The link provided is for academic-related (AR) jobs: these are staff categories that were once seen to support academic workings, but are now viewed as professional services (PS), i.e. careers in their own rights. Whether termed AR or PS, the guidelines linked to here are not for the academic (teaching and research) jobs being discussed here.
> Noted, accepted and deleted, removing the impression that this additional justification gives (i.e. that some aspects of the methodology are wrong or not reflective).

The following two points are more picky I suppose, but potentially important in under-standing the expectations of an early career academic at the SL level.

7. First of all, I don't agree with the conflation of appraisals and promotion criteria. Appraisal are designed for academics to set out aims and objectives for undertaking their job on an annual basis, and to have discussions as to whether these have been met. Promotion criteria on the other hand request for feed-back on one's reputation for mainly research and (teaching) scholarship, so to bring the two together is not particularly helpful as they work on very different timescales.

> In the manuscript there was occasionally a lack of careful distinction between the use of appraisal and promotion criteria.  Usage through the manuscript has now been modified where necessary to avoid conflating the two terms.

> Overall, in the experience of the 17 academic co-authors, the purpose of a progressive and useful appraisal system it to set aims and objectives that assist an academic to develop in their job role (i.e. working towards promotion), with both keying into universities wider objectives for obvious reasons. Thus, assessment of promotion criteria, put into the context of our experience of appraisals, provides a useful semi-objective basis (i.e. Table 2) for understanding academic motivations.

> The start of Sect 4.1.2 has been modified to lay out more clearly the rationale for our approach. P9L27

The study suggests that appraisals are "a relatively new phenomenon", but they are not; they have been undertaken at universities for over 20 years.

> "relatively new" has been removed. P9L26. We used the term 'relatively' literally. They have been used at universities for 20 years, in comparison to estimates of 50-70 in industry e.g. https://www.peoplehr.com/blog/index.php/2015/03/25/a-brief-history-of-performance-management/. Moreover, they are certainly not undertaken to "judge" (P7, line 31).

> When a grade (e.g. 'very good') influencing discretionary remuneration is given, or when at the other end this (ultimately) is underpinned by the existence of a performance management process, it is difficult to escape the view that an element of judgement is common. Text modified P9L29.

Second, the study is based around academics at the SL level, but suggests on P9, line 29 that this is equivalent to the North American Associate Professor (AP), but I would argue that AP was more equivalent to Reader, where a UK academic is recognized for their world-leading research, as would someone be in the States on being awarded tenure and AP. For a SL, teaching and scholarship plays a stronger role in evaluations.

> For clarity, in line with suggestions from R1, this international aspect has now been placed in a separate section at the end of the manuscript (Sect. 7). Also, in this academia's seniority classification is now omitted for in favour of a definition of <10 years to avoid any conflict with the main text.

8. For me, more problematic, is "de facto expected" number of hours an SL is expected to work at week, up to 50 hours or more. It describes working 10h days or working at weekends as "respectable length". These practices are greatly at odds with moves to have greater equality and work-life balance that reflects the needs for academics having caring responsibilities, to have a life outside of academia, and to minimize stress and mental health issues. The study goes on to state that the ideal person "is in good mental health" and are "efficient and effective" in their approach to research, to the point that "they would not remain in the their position if they were not" (P10, lines 1- 2). This is patently nonsense – academics are not removed from their jobs on the basis if they are efficient or not. For me this gives the impression that the model being developed - for a person to devote 0.5 days a week to collaborate with industry - will not work if an academic does not want to, or cannot work, a 50h week, or miss time with families at weekend, or experience any kind of mental health issues etc. These factors are not trivial – they have important implications for diversity and equality issues, and personally, I would question the value of such a model from the outset.

> The intention was never to give the impression that an academic cannot work with industry unless working 50+ h/week. The Environmental Scientist Persona section (Sect. 3) has been altered to remove the assertions contested by the reviewer. Indeed, if anything we sound a distinct note of caution about simply working more hours at the end of our analysis (Sect. 6.1.1) "*A convincing (self-)justification is likely therefore needed well before any official appraisal*" P19L12.

> It is outside the scope of this paper to explicitly condone, or otherwise, UK academic workloads or management practice. We have adapted the text in Sect. 6.1.1 and ensured that we remain close to the observational data.

> Neither our conceptual model (Sect. 6.2) nor recommendations (Sect 6.3) did, or now do, advocate what might be considered bad practice (e.g. working at weekends) e.g. P19L6-14.

> The model we present (Sect. 6.2) is not dependent upon, or related to expectations of length of working week. It incorporates time-pressure (Sect. 6.1.1), but the reviewer is not questioning the presence of this. How an

academic may, or may not, choose to free up 0 to 0.5 days per week is up to them; the key point for a practitioner is that some effort will have to be made to do this e.g. P25L23.

> To avoid confusion over ideal vs idealised, we have clarified (Sect. 3) that we are discussing an idealised/model/illustrative person. P6L21 We do not comment on what an "ideal" scientist may or may not be. The model persona is ours to select characteristics for, and we wish from them to remain "efficient and effective", but to account for R2's opinion we have softened their mental health to "reasonably good" and raised their demands for work life balance by removing "at least some".   P7L14, P7L20.

> We have removed the comment "they would not remain in the their position if they were not" as debating/justifying this would detract from the points in the paper.  However, we note that many universities have yearly appraisal systems (e.g. 'Performance and development review') with performance management processes associated with them. Also, see various comments in Boswell (2018).

9.  Finally, the assumption that [P13, line 20] "PhD students can be an effective means to generate publications in a time-limited university environment" is not one that I or any academic I know would condone, and has no place in modern day academia.

> This has been re-phrased P13L26.

> We admit that the brevity used here may make this sound mercenary and inappropriate. Clearly, in good practice the student's needs are paramount. But, this does not discount the clearly and frequently observed pattern that good, well-supported students produce papers; indeed they are expected to – e.g. NERC's approach to 'Research Excellence' as a means to assess DTPs relies heavily upon PhD students' publication production (https://nerc.ukri.org/about/whatwedo/engage/engagement/dtpevaluation/crac-dtp-report/
).  And, it is difficult to believe that this is not a factor in motivating scientists to spend more time supporting and engaging with PhD students than they otherwise might.

> A counter-case is provided by social science (e.g. Human geography) where co-publication is rare for cultural reasons. From anecdotal evidence talking to colleagues from various institutions, these colleagues often find it much harder to justify spending significant time with students as there is little returning benefit in terms of the key metrics by which they are assessed.

10. Are the results sufficient to support the interpretations and conclusions?

Not really. The conclusion seems to be that an efficient and effective academic can probably find 0.5days a week to collaborate with industry, but I question the data and assumptions that this is based upon. It seems to me that this figure is a qualitative amount that the academics in the cohort suggested that they may be able to find, out of their already busy schedules. Which is fine, but this does not seem to be derived from a critical analyses of what a SL does on a day to day basis.

> Please see response to comment 1 for R2 above, detailing how the manuscript has been amended to more clearly explain how the indicative figure of *up to* 0.5 days was derived (Sect 6.1.1). Based on addressing these comments, we now feel that there is a clear link between the critical analysis of the data and the conclusions of our study.

11. Do the authors give proper credit to related work and clearly indicate their own new/original contribution?

I think so.

> No response needed.

12. Does the title clearly reflect the contents of the paper?

The title mentions "business", but in the manuscript refers mainly to "industry"

> Noted, and changed to 'business' throughout. The origin of our use of 'industry' for a 'business' that does not actually make things originates in practitioners' usage typical of the term 'insurance industry' e.g. Dixon *et al* [2017].

13. Does the abstract provide a concise and complete summary?

This was fine.

> Thank you.

14. Is the overall presentation well structured and clear?

I thought that the Introduction, setting the scene etc could be much more succinct.

> The Introduction is now shorter, focussing only on framing the academic question.

Many times the manuscript refers the reader to future sections as justification of what is being stated, but this made reading of the paper difficult, as you have to keep going backwards and forwards to find out what is being referred to.

> Forward referencing of sections has now been minimised, for instance within in '*Environmental Scientist Persona*', which has been moved forwards in the manuscript to assist this. Forward referencing is retained where its purpose is to clarify the upcoming structure of the paper (e.g. in the introduction), or where arguments are necessarily interconnected within the discussion.

15. Is the language fluent and precise?

Overall it is fine. But there were a few tropes that could be avoided, such as P17, lines 2-3 "So, this work confirms that an academic with days and days to sit gazing around and pondering is a myth. . .". This suggests a misunderstanding of how an academic may approach their writing, rather than just doing it. Writing does require thinking and pondering, for days, sometimes weeks or months, so I'm not really sure what is being got at here.

> Accepted. This phrase originates in an earlier version of this work intended to communicate to stakeholders, adding 'colour' and directly counteracting potential stereotypes about academics having 'spare' or 'free' time. This is now stated directly. P17L20

16. Are the number and quality of references appropriate?

These look fine.

> Thank you.

**Reviewer 3 – Dr James Esson (Loughborough i.e. 'internal')**

*1. Abstract/Summary/Introduction*

1a) Originality

The language could be more assertive in parts to better signal the original aspects of the study, particularly re the conceptual framework ('tentatively posit' is overly cautious).

> Changed to 'propose'. P2L9.

The abstract does indicate that the study involved an original empirical data set, but it is not clear what if anything is innovative about the approach – this would be a nice addition but difficult to do given word constraints.

> This innovative aspect of the work was (and is) flagged at the end of the Introduction, but is now highlighted more strongly in the Methods section. It is not included in the abstract to retain the flow of the text and clarity of the main messages, however it is now included the conclusions (P32L25).

1b) Significance

The aim and implications of the paper are clear in the abstract and intro.

> Thank you. Not action needed.

The opening sentence needs refining in the former, but the message is broadly on point.

> We believe the opening sentence in the abstract is suitable, and have not altered it.

As mentioned above, the development of a new conceptual model and the implications of the findings are mentioned in quite tentative language in the abstract. I would encourage the author to be a little bolder about the originality and significance of the framework.

> The language in the abstract is now less tentative (e.g. P24L21-23), and the utility of the model w.r.t the practical suggestion is now more clearly signposted in Sect 6.2.3. The novelty and utility of the new conceptual model are now also raised in the conclusions (P33L16-18).

Also, more could be said about what people will/should do differently as a result of this research (abstract).

> The last line of the abstract has been amended to indicate where behaviour changes may arise. P2L11

1c) General point – The introduction is too long and should perhaps be separated to make two sections (intro and literature review)

> The Introduction has been divided into '*Introduction*' and '*Case study: Insurance sector*'. P4L1

*2. Research Design/Methods*

2a) Originality
The study adopts a mixed-methodology informed by an action research approach (content analysis, interviews and workshop discussion). Given that the author demonstrates the originality of the research in the introduction, the novelty of the research design is downplayed somewhat, and 'pragmatism' and time efficiency are even stated before effectiveness as reasons for the approach taken. This might be true, and most academics have no doubt been in a similar position, but I would not encourage the author to emphasis this. Rather, it would be better to draw attention to the alignment between the aims, methodology and methods used, and the innovative elements of this approach.
> Methods are now described in the Methods section by more directly aligning them to project aims (P7L31), and this is briefly noted in the Conclusions.

2b) Significance
If the paper achieves the impact the authors anticipate, it will become primary or essential point of reference in this area. As indicated above, the current tone does not reflect this. Or put differently, the significance of the research design is downplayed which is odd given that some aspects are highly novel (e.g. the Environmental Scientist Persona).
> The highly novel research methodology is now downplayed less in the Abstract, Methods and Conclusions (see comment immediately above).

2c) Rigor
The research design and techniques of investigation and analysis are explained clearly and thoroughly. Particularly attention is paid to ethical concerns, potential limitations and biases. This is done sensibly, in that it demonstrates integrity and rigour, while avoiding raising concerns about the appropriateness of the research design and validity of associated findings.
> Thank you. No response necessary.

*3. Discussion/Conclusions*

3a) Originality
Yes, the findings of the research and the wider implications are clearly stated. The discussion provides a new conceptual framework and associated recommendations to inform practice.
> Thank you. No response necessary.

3b) Significance
A strong case is made as to why the article enhances understanding and/or practice, and the implications of this for university-business collaborations in STEM subjects as well as the social sciences.
> Thank you. No response necessary. See P24L27-30
> The case for the global reach of these implications could be qualified slightly to better reflect the Euro, North America and Australian focus of the article. A case could be made that these issues are likely to, or are already, occurring in what we could call 'emerging economies'.
> The case of international applicability is now in its own section (Sect. 7), and the reviewers suggest of a distinction between developed and emerging economies has been incorporated. [Also see R1C2 and R2C2].

3c) Rigor
There is clear intellectual coherence and alignment between the aims, methodology and the findings.
> Thank you. No change needed.
I would recommend having a few more citations in the conclusion to support the more general claims being made e.g. ''It is well established that most scientists are driven by curiosity not additional personal financial reward'' (see also XXX; XXX).
> In the style we adopt, few citations are needed within Conclusions, but some have been added where appropriate.

[revised manuscript text omitted]

John Hillier 12/11/18 17:24
**Comment [24]:** R1C17

John Hillier 15/11/18 16:42
**Comment [25]:** R1C18, R2C15

John Hillier 26/11/18 16:46
**Comment [26]:** R2C1 – Clarification of the origin of the figure of 0.5 days/week in the two paragraphs below, so that transparency allows readers to form a better view on the uncertainty within this figure.

might not immediately be thought of by business practitioners when considering an academic doing research. Each of these 10 illustrative main duties is multi-faceted. For instance, 'own hands-on research' conducted by the academic includes elements such as reading articles, modelling, programming, learning any new skills required, and writing journal articles. We do not argue that this categorisation is the only one possible, or that the tasks itemised are prescriptive. Indeed, a number of alternative tasks that could be prioritised and substituted in for any individual academic are reported in Sect. 5.2. However, the view of the 17 academic co-authors and workshop participants is that Table 1, distilled from job specifications, is on balance a fair representation of the demands on a UK academic. To wit, key elements are present and the number and magnitude of tasks form a suitable basis for an evidence-based, indicative view of time that might be available for impact-based work.

[Figure]

Fig. 3: Potential time availability for collaboration with business, in the context of other duties, of a typical early- to mid-career UK academic. Time available is divided between Teaching, Research and Administration 2:2:1 (see main text), and then an academic's own hands-on research, writing and impact-related work are only 3 of 10 tasks within 'Research', giving 0.2 days/week for each task assuming an equal distribution.

Fig. 3 summarises the logic behind an estimate of *up to* 0.5 days/week. A ratio of 2:2:1 for R:T:L/A leads to 2-days per week for 'Research'. Then, an equal distribution of time between the 10 tasks within this category implies ~0.2 days/week for impact-based work. This is moderated by the knowledge and experience of the 17 academic co-authors, assimilating the relative priority that must be given to the tasks in light of assessment criteria (Sect. 5.3). Without some special circumstances to buy out an academic's time (e.g. KE Fellowship), for some co-authors 1 h was a limit, and not even that in term time. The experience of others is that, with determination, it is possible to preserve 1 day/week for the totality of the three tasks related to hands-on research (see Fig. 3). Ultimately, the co-authors' consensus is that, if strongly prioritized (i.e. intermediate term benefits clearly identified), 0.5 days/week was a ceiling to what *might* be possible. This said, readers can review the evidence (e.g. Table 1, Fig. 3) and form their own view.

John Hillier 14/11/18 20:30
**Comment [27]:** Figure to better explain derivation of <0.5 days/week R2C1. Figure suggested by R1C4.

[revised manuscript text omitted]

John Hillier 14/11/18 21:14
**Comment [46]:** R1C25. Rephrased.

John Hillier 13/11/18 15:56
**Comment [47]:** R1C1 – PhD comment replaced.

John Hillier 15/11/18 23:00
**Comment [48]:** R2C1 – moved PhD comment and added UKRI initiatives.

- *Provide access to training, expertise (e.g. actuaries) or networks* - primarily a mechanism to maintain contact and alignment, since academics are typically proficient at obtaining these already.

Although apparently a counterpoint to the main theme of this article, aimed at risk practitioners, an illustrative list of actions a university scientist may take to support their risk practitioner is given below; it may assist practitioners new to the role of collaboration with academics or as an aid to give to an academic new to collaborating with insurers.

In this spirit, it is worth giving a precis of motivations within this industrial sector. As individuals, it is notable that practitioner's motives are mixed, with curiosity (i.e. the 'puzzle') and family common drivers, not just the 'gold' (see Sect. 6.2, (Lam, 2011)). Whilst insurers ultimately require increased profitability, and approaches to quantify this to create a business case for collaboration are mixed and varied, three main routes exist; training, operational utility (e.g. data, tools), or reputational enhancement. The latter works by differentiating the company from its competitors (i.e. more accurate risk assessment through better science), providing arguments for retaining existing clients, and opening doors to new clients that sales teams can follow up. In (re)insurance this can be more important than harvesting and protecting IP generated in collaborations.

The following list is of ways research scientists might provide support to their risk practitioner partner. These are mapped to the typology of impact (i.e. practitioner benefit) in Sect. 6.2.2 using square brackets e.g. [2] and include a brief commentary on how and why benefits emerge. Suggestions are not ranked as utility will be case specific.

- *Undertake a literature review* e.g. comprehensive review of what is known about risks in an emerging peril-region such as Africa [4]. This is a safe (i.e. low risk), early stage deliverable if included into funding bids. It will appear least like a burden to the academic if the subject is novel (i.e. publishable) and a likely impact (e.g. pending strategic decision) has been identified. It is time-efficient for the practitioner.
- *Deliver new research-based science* in the form of concepts or theories that can be implemented by the practitioner to operational advantage ahead of competitors, e.g. by engaging with the scientist in a co-designed project as the work progresses [1]. Feed-in could be by modifying a company's 'own view of risk', or by some adaption to their natural hazard risk process/model (e.g. catastrophe modelling). When exploring ideas or methodological improvements at the cutting-edge (i.e. higher-risk), collaboration can be a low cost alternative for a practitioner as if sufficient novelty exists a substantial fraction of the cost might be supportable through public funding.
- *Develop a spreadsheet-based* 'decision support tool' [1]. Although this is too basic for most (re)insurance users, it may be appropriate for some of their clients.
- *Provide training sessions* [3]. See list above.

John Hillier 14/11/18 22:10
**Comment [49]:** R1C8 – Before publication, we intend to add a link to an upcoming briefing note 'Funding Science for Natural Hazards Insurance' that we are currently finalizing.

John Hillier 13/11/18 15:18
**Comment [50]:** R1C26
John Hillier 21/11/18 13:01
**Comment [51]:** R1C6

[revised manuscript text omitted]